# Beyond a PPR-RNA recognition code: Many aspects matter for the multi-targeting properties of RNA editing factor PPR56

Yingying Yang, Kira Ritzenhofen, Jessica Otrzonsek, Jingchan Xie, Mareike Schallenberg-Rüdinger *, Volker Knoop *

IZMB–Institut für Zelluläre und Molekulare Botanik, Abteilung Molekulare Evolution, Universität Bonn, Bonn, Germany

* mareike.ruedinger@uni-bonn.de (MS-R); volker.knoop@uni-bonn.de (VK)

## Abstract

The mitochondrial C-to-U RNA editing factor PPR56 of the moss *Physcomitrium patens* is an RNA-binding pentatricopeptide repeat protein equipped with a terminal DYW-type cytidine deaminase domain. Transferred into *Escherichia coli*, PPR56 works faithfully on its two native RNA editing targets, nad3eU230SL and nad4eU272SL, and also converts cytidines into uridines at over 100 off-targets in the bacterial transcriptome. Accordingly, PPR56 is attractive for detailed mechanistic studies in the heterologous bacterial setup, allowing for scoring differential RNA editing activities of many target and protein variants in reasonable time. Here, we report (i) on the effects of numerous individual and combined PPR56 protein and target modifications, (ii) on the spectrum of off-target C-to-U editing in the bacterial background transcriptome for PPR56 and two variants engineered for target re-direction and (iii) on combinations of targets in tandem or separately at the 5'- and 3'-ends of large mRNAs. The latter experimentation finds enhancement of RNA editing at weak targets in many cases, including cox3eU290SF as a new candidate mitogenome target. We conclude that C-to-U RNA editing can be much enhanced by transcript features also outside the region ultimately targeted by PPRs of a plant editing factor, possibly facilitated by its enrichment or scanning along transcripts.

## Author summary

RNA-binding pentatricopeptide repeat (PPR) proteins convert specific cytidines to uridines in mitochondrial and plastid transcripts of land plants to correct genetic information. PPR protein PPR56 is assigned to two editing sites in *nad3* and *nad4* transcripts of the moss *Physcomitrium patens*. Transferred into *Escherichia coli*, PPR56 edits its co-delivered native targets, but also more than 100 specific cytidines in endogenous bacterial transcripts. We have used the *E. coli* editing system to intensively study the editing properties of PPR56 as a prime example of a plant-type C-to-U RNA editing factor. We confirm that the selection of editing sites depends on the PPR region of the editing factor, which recognizes the RNA target upstream of the C to be edited. Some single amino acid

**Data Availability Statement:** The authors confirm that all data underlying the findings are fully available without restriction. The RNAseq data have been deposited in the SRA archive as BioProject PRJNA984633. All other relevant data are within

the manuscript and its Supporting Information files.

**Funding:** The author(s) received no specific funding for this work.

**Competing interests:** The authors have declared that no competing interests exist.

modifications within this region can re-direct the editing factor to new targets, but others reduce or prevent editing completely. Modifications on the target side even outside of the RNA region recognized by the editing factor also affect editing efficiencies significantly. The combination of targets, for example, enhances the editing of the weaker target in many cases. Thus, not only the direct target-protein interaction, but also other transcript features influence the final selection of editing sites.

## Introduction

The recent years have seen much progress towards understanding the molecular machinery behind cytidine-to-uridine RNA editing in plant chloroplasts and mitochondria [1–4]. Among other insights, very early functional studies on plant RNA editing based on *in organello*, *in vitro* or transplastomic studies had already demonstrated that the specificity for identifying cytidine targets largely resides in their immediate sequence environment, mainly within circa 20 upstream nucleotides [5–14]. The molecular characterization of the *trans*-acting specificity factors, however, ultimately relied on reverse genetic approaches leading to the identification of CRR4 as a first identified chloroplast and MEF1 as the first mitochondrial RNA editing factor in *Arabidopsis thaliana* [15,16]. Such site-specific editing factors feature sequence-specific RNA-binding pentatricopeptide repeats (PPRs) of the so-called PLS-type followed by "extension" domains E1 and E2 plus a C-terminal DYW cytidine deaminase, which may alternatively be supplied *in trans*. PLS-type PPR arrays typical for RNA editing factors are characterized by "long" and "short" PPR variants with characteristic consensus profiles along with the canonical P-type PPRs of 35 amino acids. The E1 and E2 motifs of 34 amino acids are distantly related to TPRs (Tetratricopeptide Repeats) existing in other proteins, where they have been shown to mediate protein-protein rather than protein-RNA interactions. The more recent research on RNA editing and other processes of RNA maturation in the two endosymbiotic organelles of plant cells has clearly profited from parallel approaches taken not only with model flowering plants like *Arabidopsis*, maize or rice but also with bryophyte model organisms [17]. Flowering plants (angiosperms) feature complex RNA editosomes variably composed of numerous and diversely interacting proteins to target specific sites for C-to-U conversion in the organelle transcriptomes. Aside from the core RNA editing factors for target recognition, the complex angiosperm editosome protein assemblies, for example, feature MORFs/RIPs or ORRMs [3,18–20].

In contrast, a much simpler scenario has emerged for C-to-U RNA editing in "early-branching" land plants among which the moss *Physcomitrium patens* holds a key role as a model organism [4,21,22]. All characterized RNA editing factors in *Physcomitrium* combine a stretch of pentatricopeptide repeats (PPRs) responsible for sequence-specific RNA recognition with a terminal DYW-type cytidine deaminase carrying out the site-specific C-to-U conversion.

To a large part, the complex editosomes of angiosperms seem to be the result of frequent separation of RNA target recognition and the catalytic DYW domain, now relying on protein-protein interaction including various helper proteins interacting *in trans* [23–31]. This evolutionary pathway is exemplified with the recently investigated case of angiosperm RNA editing factor CWM1 that is C-terminally truncated in *Arabidopsis* and relies on helper proteins but features an orthologue with a terminal DYW domain in the early-branching flowering plant *Macadamia* that was able to complement an RNA editing KO in *Physcomitrium* [32]. Single

editing factors retaining those functionalities in just one polypeptide, as in the case of the here investigated PPR56, mainly exist in early-arising plant lineages like the mosses [3].

*Physcomitrium patens* has a prominent role with its only 13 C-to-U RNA editing sites assigned to nine site-specific RNA editing factors [3,21,22,33]. However, *Physcomitrium* is in no way representative for other bryophytes, which feature the full spectrum of RNA editing being entirely absent in the marchantiid liverworts, with massive C-to-U RNA editing in the early-branching moss *Takakia lepidozioides* [34] or with abundant "reverse" U-to-C RNA editing co-existing with C-to-U editing in hornworts like *Anthoceros agrestis* [35]. Among altogether more than 100 pentatricopeptide repeat proteins in *Physcomitrium* only nine are RNA editing factors and all of them, including PPR56 investigated here, are characterized by a PLS-type PPR array linked to a terminal DYW cytidine deaminase domain via the E1 and E2 domains [21].

It is likely no surprise that the simple one-protein RNA editing setup of *Physcomitrium* could be functionally transferred into heterologous systems like the bacterium *Escherichia coli* [36] and, more recently, also into human cell lines [37]. The bacterial setup in particular offers an easy access to exploring the interaction of an RNA editing factor and its targets by allowing the investigation of numerous protein and target variants in short time.

The mitochondrial RNA editing factor PPR56 of *Physcomitrium patens* has been functionally characterized some years ago [38] and appeared particularly suited for further investigations for several reasons. Firstly, it has two native mitochondrial target sites that are converted with different efficiencies by specific cytidine deamination in the moss (Fig 1A). Editing target nad4eU272SL is converted to more than 99% in the steady state mitochondrial transcriptome of *Physcomitrium*. Editing efficiency at its second target, nad3eU230SL, is more variable and may depend on environmental conditions but is generally above 70% *in planta* [38,39]. The RNA editing target site labels follow a nomenclature proposal that indicates the respective genetic locus (here *nad* subunits of respiratory chain complex 1, the NADH ubiquinone oxidoreductase), the RNA editing event towards uridine (eU), the transcript position counting from the first nucleotide of the AUG start codon and the resulting codon change, here serine to leucine in both cases [39,40].

Defining a PPR-RNA recognition code has been a tremendous step forward in understanding the operation of pentatricopeptide repeat proteins [41–44]. At the core of this code, the identities of the 5th and the last (L) amino acid within the two antiparallel α-helices constituting an individual PPR are key to recognizing individual ribonucleotides with position '5' distinguishing purines (adenosines or guanosines) from pyrimidines (cytidines or uridines) and position 'L' defining preferences for amino (A or C) or keto nucleobases (G or U). However, the situation is notably more complex for PPR proteins acting as RNA editing factors, which not only feature canonical 'P-type' PPRs of 35 amino acids but also variants with different consensus profiles and slightly variable lengths. Most widely distributed are the variants 'L' (long, 35–36 aa) and 'S' (short, 31–32 aa) contributing to PLS-type PPR arrays in most plant RNA editing factors. Yet more PPR variants such as 'SS' and 'LL' have recently been identified in the growing amount of genomic data for the huge PPR gene families in land plants, now also including hornworts, lycophytes and ferns [45].

The PPR-RNA code outlined above can be applied only to P- and S-type but not to L-type PPRs and the functional role of the latter remained mysterious. Notably, despite a conceptually slightly better overall fit of the nad3eU230SL target to the P- and S-type PPRs of PPR56 (Fig 1A), the nad4eU272SL target is edited more efficiently not only in the native moss background but also in the recently established heterologous *E. coli* RNA editing assay system [36]. Hence, additional parameters beyond the conceptual matches of an array of PPRs to its targets evidently contribute to RNA editing efficiencies.

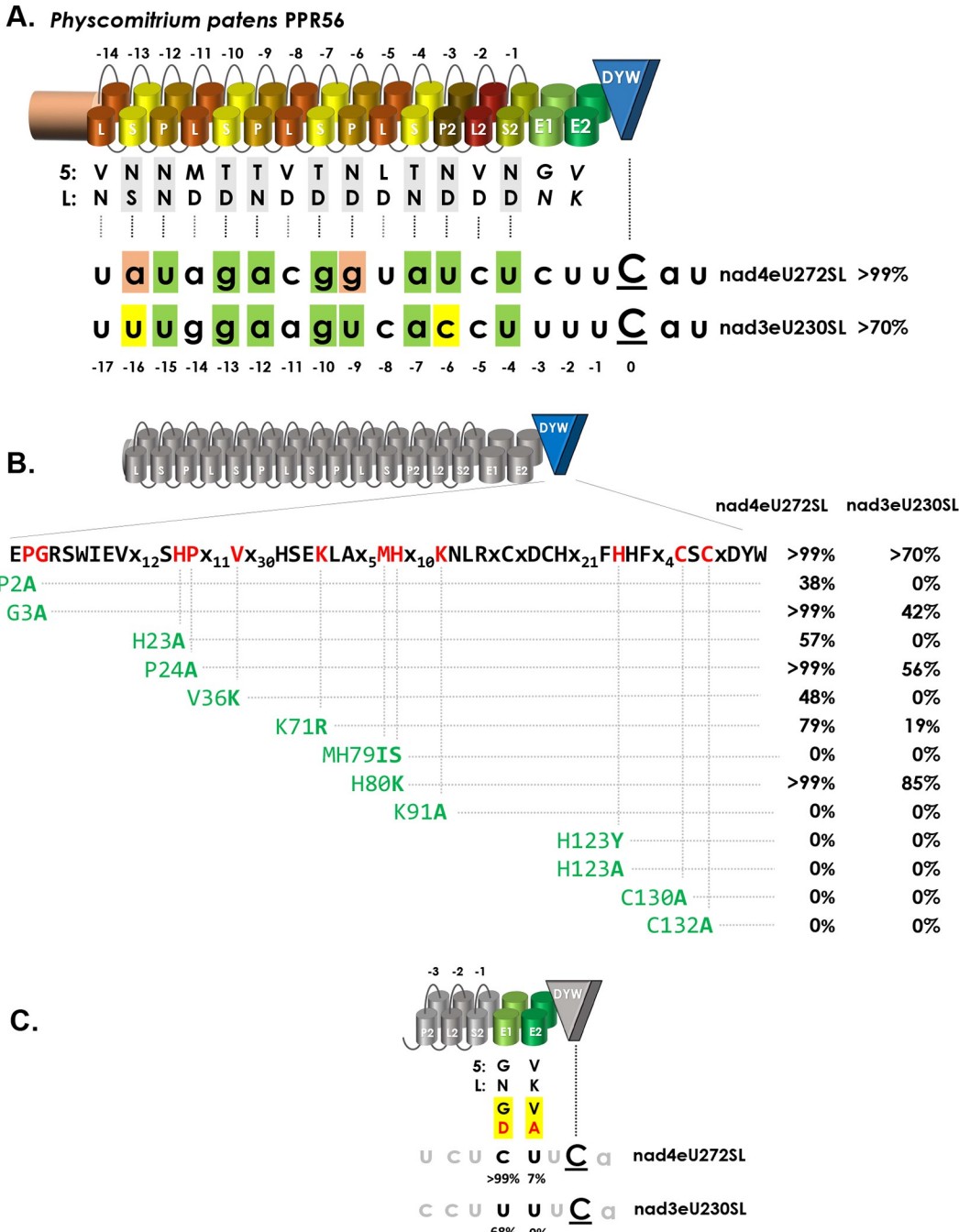

**Fig 1. PPR56 and site-directed mutations in its DYW cytidine deaminase domain. A.** Makeup of PPR56 and its two native targets. PPR56 is a typical plant organelle RNA editing factor featuring a PLS-type PPR array with alternating P-, L- and S-type PPRs followed by extension motifs E1 and E2 and a terminal DYW cytidine deaminase domain. Typically, the most C-terminal PLS triplet of plant editing factors has a deviating consensus and is labeled P2-L2-S2. As suggested previously [88], to account for generally more loosely conserved N-terminal repeats, PPRs are numbered backwards with the terminal PPR S2-1 juxtaposed with position -4 upstream of the editing target cytidine converted into uridine. Shading of matches in green follows the PPR-RNA recognition code based on amino acid identities in positions 5 and L in P- and S-type PPRs: T/S+N:A, T/S+D:G, N+D:U, N+S:C, N+N:Y. The corresponding amino acid identities in the TPR-like E1 and E2 motifs are indicated in italics. PPR56 has two native editing targets in the mitochondria of *Physcomitrium patens*: nad4eU272SL and nad3eU230SL. Near-complete editing (>99%) is generally observed for the nad4eU272SL target, but lower editing (>70%) is variably observed for nad3eU230SL *in planta*, possibly as a result of different strains or cultivation conditions [38,39]. **B.** Mutations in the DYW domain of PPR56. Twelve conserved amino acid positions (see S1 Fig) in the DYW domain of PPR56 were selected for mutations and tested on both native targets nad4eU272SL and nad3eU230SL in

the *E. coli* RNA editing assay system. RNA editing efficiencies are given as the mean of at least three biological replicates (independent primary *E. coli* clones) when RNA editing activity was detected. Initially identified absence of RNA editing for a construct was confirmed with at least one additional independent bacterial clone. All primary data for RNA editing assays are given in S1 Data. **C**. Mutations in the E1 and E2 motifs of PPR56. Positions 34 ('last') of the E1 and E2 motifs potentially juxtaposed with nucleotide positions -3 and -2 upstream of the edited cytidine have been mutated and tested on the two native targets of PPR56 with RNA editing remaining unaffected by the PPR56|E1:N34D mutant, but dropping dramatically for the PPR56|E2:K34A mutant.

Here, we explored the impact of PPR56 protein mutations and of modified, extended, combined and differently placed RNA targets in the easily amenable bacterial system to identify the relevant elements contributing to efficient RNA editing. Most importantly, we found that sequences further upstream of the region ultimately bound by the PPR array contribute to high RNA editing efficiency and that tandem combinations of target sequences can significantly enhance RNA editing at previously less efficiently edited downstream targets. The latter include both selected off-targets in the *E. coli* transcriptome as well as cox3eU290SF as a predicted further candidate plant mitogenomic target of PPR56.

Moreover, we observed that placing the otherwise moderately edited nad3eU230SL target of PPR56 in the 5'- vs. the 3'-UTR of a long mRNA can enhance RNA editing even above the level observed in its native plant mitochondrial environment. Hence, the wider environment of the core RNA target sequence as defined by the PPR array contributes notably to the observed RNA editing efficiencies. Altogether, we conclude that the operation of PLS-type RNA editing factors like PPR56 relies not only on the defined code for P- and S-type PPRs but also on the hitherto enigmatic L-type PPRs and on the wider transcript environment.

## Results

### PPR56, mutant nomenclature and the vector assay systems

PPR56 is a plant C-to-U RNA editing factor equipped with a highly conserved carboxyterminal DYW-type cytidine deaminase domain linked to an upstream PLS-type PPR array via the E1 and E2 extension motifs (Fig 1A). For clarity, we here introduce nomenclature standards to label mutations on the protein or on the target side, respectively, that have been introduced for studying RNA editing functionality. For mutations on the protein side, we use a protein domain label behind a pipe symbol, followed by a colon and the position and amino acid identities in single-letter annotation before and after changes, e.g. PPR56|DYW:G3A for the mutation converting the glycine of the conserved PG box (Figs 1B and S1) into alanine. As a shorthand notation for mutations targeting the crucial positions '5' and 'L' of a given PPR, we simply indicate the introduced identities without numbering, e.g. PPR56|P-6ND>TD for the mutation converting the native ND combination in PPR P-6 for a conceptually better match to the guanidine that is naturally present in position -9 upstream of the nad4eU272SL editing site (Fig 1A).

For mutations on the RNA target side, we will use small letters to label nucleotide changes and indicate positions relative to the editing site, which are added behind the respective RNA editing site labels after pipe symbols. For example, nad4eU272SL|u-4g will indicate the U-to-G exchange introduced four nucleotides upstream of the RNA editing site, which is assumed to be juxtaposed with the terminal S2-type PPR of PPR56 (Fig 1A).

We mainly used the previously established heterologous expression system in *Escherichia coli* based on vector pET41Kmod [36]. The coding sequence of PPR56 is cloned in-frame to an upstream His$_6$-MBP tag behind an IPTG-inducible T7 promoter controlled by the lac operator and the respective target sequences are inserted in the 3'-UTR followed by a T7 terminator sequence. For further experimentation allowing to place target sequences alternatively also in

the 5'-UTR, we equipped pET41Kmod with an additional MCS upstream of the protein coding sequence, giving rise to pET41Kmod2 (S2 Fig).

## Mutating the DYW domain and upstream E motifs

Mutations had previously been introduced into the DYW domain of PPR65, another *Physcomitrium patens* RNA editing factor, to confirm the crucial role of conserved amino acids residues, including the ligands of a $Zn^{2+}$ ion in the catalytic center of the cytidine deaminase [36]. Here, we have focused on other evolutionarily conserved positions in the DYW cytidine deaminase domain of PPR56 (S1 Fig). Introducing mutations into the DYW domain of PPR56 (Fig 1B) has the advantage that effects can be tested on its two native targets in parallel as opposed to only one target in the case of PPR65. The new set of mutants now also addresses a second Zn-binding site at the C-terminus of the DYW domain suggested to play a structural role outside of the catalytic center [46–48]. All mutations eliminating the relevant histidine or cysteine residues for coordination of the second zinc (PPR56|DYW:H123A, H123Y, C130A and C132A) indeed fully abolished detectable RNA editing on both targets (Fig 1B).

Other mutations further upstream in the DYW domain, however, had surprisingly differential effects on the two targets of PPR56 with a generally much stronger impact on the less efficiently edited *nad3* target, which turned out to be generally more sensitive also upon other alterations (see below). Replacing proline with alanine in the eponymous PG box at the N-terminus of the DYW domain (PPR56|DYW:P2A) has a much stronger effect than the corresponding replacement of the following glycine residue (G3A), despite 100% conservation of the latter in all nine *Physcomitrium* RNA editing factors (S1 Fig). Similarly, despite universal conservation of a downstream HP dipeptide motif in all *Physcomitrum* RNA editing factors (S1 Fig), the corresponding mutations PPR56|DYW:H23A and P24A show significant remaining RNA editing activity with the exception of H23A on the *nad3* target (Fig 1B). The position directly following the glutamate E70 in the catalytic center is conserved as either lysine or arginine in the DYW domains of RNA editing factors (S1 Fig). However, exchanging lysine against arginine in that position (PPR56|DYW:K71R) results in significantly reduced RNA editing of 79% at the *nad4* and of only 19% at the *nad3* target, respectively (Fig 1B). Notably, the reverse exchange (PPR65|DYW:R71K) had similarly led to reduced editing efficiency for PPR65 [36], indicating that the respective identity of the basic amino acid in this position is more important than could be expected.

We also addressed a variable region in the DYW domain that was previously postulated to confer compatibility for creation of editing factor chimeras [49]. Exchanging the MH dipeptide to IS (MH79IS) abolished editing activity completely whereas the single amino acid exchange (H80K) had no negative, but even a slightly enhancing effect on the nad3eU230SL target (Fig 1B). The crystallization study of the DYW domain of OTP86, a chloroplast RNA editing factor of *Arabidopsis thaliana*, suggested a regulation mechanism for DYW-type cytidine deaminases and defined a "gating domain" blocking the catalytic site in an inactive state [46]. We tested the function of the corresponding region in PPR56 by changing a conserved hydrophobic residue in its center into a positively charged lysine (V36K), which abolished editing of the *nad3* target completely and reduced editing of the *nad4* target to 58% (Fig 1B). The lysine in position 91 was found to mediate the accessibility of the catalytically important E70 of the OTP86 DYW cytidine deaminase and exchanging the K in this position in PPR56 to A (K91A) abolishes editing activity on both targets altogether (Fig 1B).

The precise role of the TPR (tetratricopeptide repeat)-like E1 and E2 motifs linking the DYW domain and the upstream PPR arrays of plant RNA editing factors is still unclear. Given their location in the proteins and the distant similarity of TPRs to PPRs, E1 and E2 may

contribute to binding to nucleotide positions -3 and -2 upstream of the cytidine target, but with a matching code different from the one for the P- and S-type PPRs. In the E2 motifs of RNA editing factors, the corresponding positions '5' and 'L' (i.e. position 34 in the TPR-like E motifs) are strongly dominated by valine (V) and lysine (K), respectively, suggesting a structural rather than nucleotide-specific role. In contrast, position '5' of E1 shows no significant conservation at all, but position 'L' (34) of the E1 motif has a resemblance to PPRs with aspartate (D) or asparagine (N) dominating in the conservation profiles. Given the unique opportunity to test impacts on the two different targets of PPR56 we created mutants PPR56|E1:N34D and PPR56|E2:K34A and tested them on both targets (Fig 1C). While mutant PPR56|E1:N34D could be expected to now favor the *nad3* target given presence of a uridine in position -3, we did not see a significant change of editing efficiency at either target. In stark contrast, editing efficiencies dropped dramatically for mutant PPR56|E2:K34A to only 7% on the nad4eU272SL target and abolished RNA editing altogether at the nad3eU230SL target, again confirming the overall higher sensitivity of the latter.

## Mutations in target positions juxtaposed with P- and S-type PPRs

To explore the different efficiencies of RNA editing at the two native targets of PPR56, we first extended the set of mutations in target positions juxtaposed with the P- and S-type PPRs that are assumed to follow the known PPR-RNA code rules (Fig 2). Only one target mutation had previously been found to enhance RNA editing at the *nad3* target: nad3eU230SL|c-6u, which improves the conceptual fit to PPR P-3ND, hence fitting expectations. In the majority of mutants, we observe that effects are much stronger for the nad3eU230SL than for the nad4eU272SL target (Fig 2). Examples are nad4eU272SL|u-4c (63%) vs. nad3eU230SL|u-4c (0%), nad4eU272SL|a-7g (20%) vs. nad3eU230SL|a-7g (0%), nad4eU272SL|g-10a (27%) vs. nad3eU230SL|g-10a (0%), nad4eU272SL|g-13a (35%) vs. nad3eU230SL|g-13a (0%) and, most dramatically for nad4eU272SL|u-15c (>99%) vs. nad3eU230SL|u-15c (0%). The latter case is particularly surprising given that (i) N-terminal PPRs generally play minor roles, (ii) PPR P-12NN is not expected to discriminate between U and C and (iii) both natural targets have a uridine in that position.

A cytidine would be expected to best match PPR S-13NS. Accordingly, we also included mutants nad4eU272SL|a-16c and nad3eU230SL|u-16c and the double target mutant nad3eU230SL|u-16c|c-6u in our collection of mutant variants. For the conceptual improvement of the matches opposite of PPR S-13NS we do observe moderate decreases in editing efficiencies to 77% for the nad4eU272SL target and to 68% for nad3eU230SL, in line with the finding that N-terminal PPRs of editing frequently do not match their target nucleotides following the PPR-RNA code. However, the double target mutant nad3eU230SL|u-16c|c-6u adding the favorable c-6u exchange still performed very well with an editing efficiency of 93% (Fig 2).

Exchanging conceptually perfect matches to PPRs P-9TN and S2-1ND through mutations a-12g or u-4g abolishes RNA editing at both targets alike, again fitting expectations (Fig 2). Combining deleterious mutations g-13a and g-10a abolishes editing not only at the *nad3* target but also at the *nad4* target completely, indicating an additive effect (Fig 2). Changing the positions where the two targets differ opposite of P- or S-type PPRs to the respective other nucleotide identities reduced RNA editing in both cases, to 54% for nad4eU272SL|a-16u|g-9u|u-6c and to 49% for nad3eU230SL|u-16a|u-9g|c-6u, respectively.

## Mutants in the PPR array

We tested whether target sequence mutations could be compensated by protein mutations in the corresponding PPRs (Fig 3). This was not the case for nad4eU272SL|u-4c, edited to 63% by

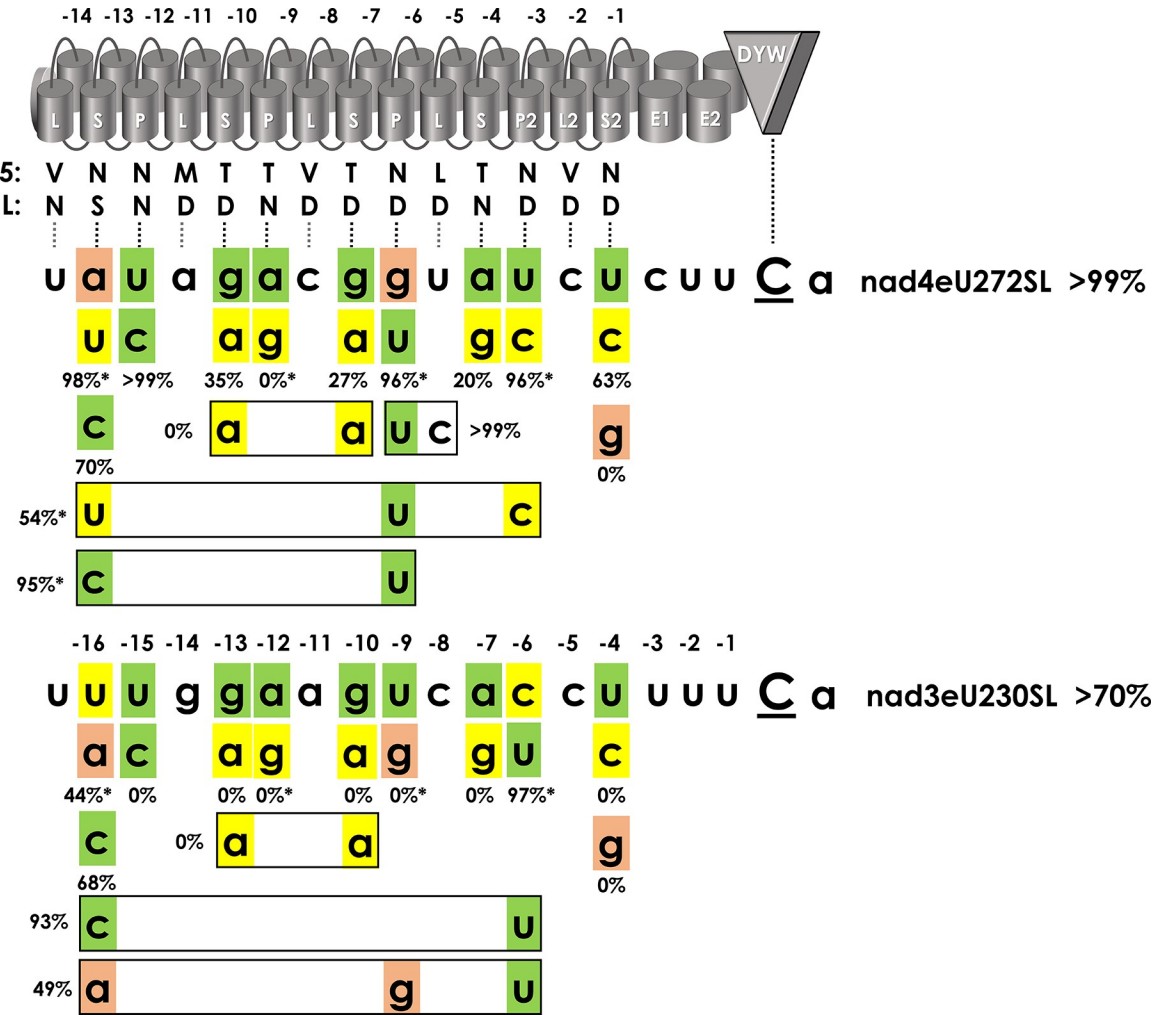

**Fig 2. PPR56 target mutations opposite of P- and S-type PPRs.** Mutations have been introduced upstream of the two native PPR56 editing targets nad4eU272SL and nad3eU230SL in positions juxtaposed with P- and S-type PPRs assumed to follow the PPR-RNA code rules for amino acid positions 5 and L. Ten target mutants investigated earlier [36] are indicated with asterisks at the respective percentages (e.g. for nad4eU272SL|a-16u, top left). Designation of PPRs, numbering of positions and shading in target sequences is as in Fig 1A. Average RNA editing activities from three replicates are given below individually mutated positions or next to multiple mutations (boxed). Primary data are listed in S1 Data.

unmodified PPR56 (Fig 2), but to only 30% by the conceptually adapted version PPR56|S2-1ND>NS (Fig 3A). Moreover, target variant nad3eU230SL|u-4c was neither edited by PPR56 (Fig 2) nor by PPR56|S2-1ND>NS (Fig 3A). Unmodified targets nad4eU272SL and nad3eU230SL were still edited to 78% and 27% by the modified PPR56, respectively. Notably, canonical positions 5 and L in the terminal S2-1 PPR matching with the corresponding position -4 as in PPR56 are more of an exception than the rule for plant RNA editing factors.

For five other mutations in specific PPRs (S-13NS>ND, P-12 NN>NS, S-7TD>TN, P2-3ND>NS and L2-2VD>ND, respectively), we found that RNA editing of the native targets was likewise significantly decreased (with the exception of PPR56|P-12NN>NS on the *nad4* target) and could not be rescued by corresponding mutations in either target (Fig 3A). This is most prominently seen for S-7TD>TN abolishing RNA editing altogether and which could not rescue the corresponding mutation g-10a (Fig 3A). Other mutations in the P2-L2-S2 triplet, again, had generally stronger effects on the *nad3* target. Adapting P2-3 for a conceptually

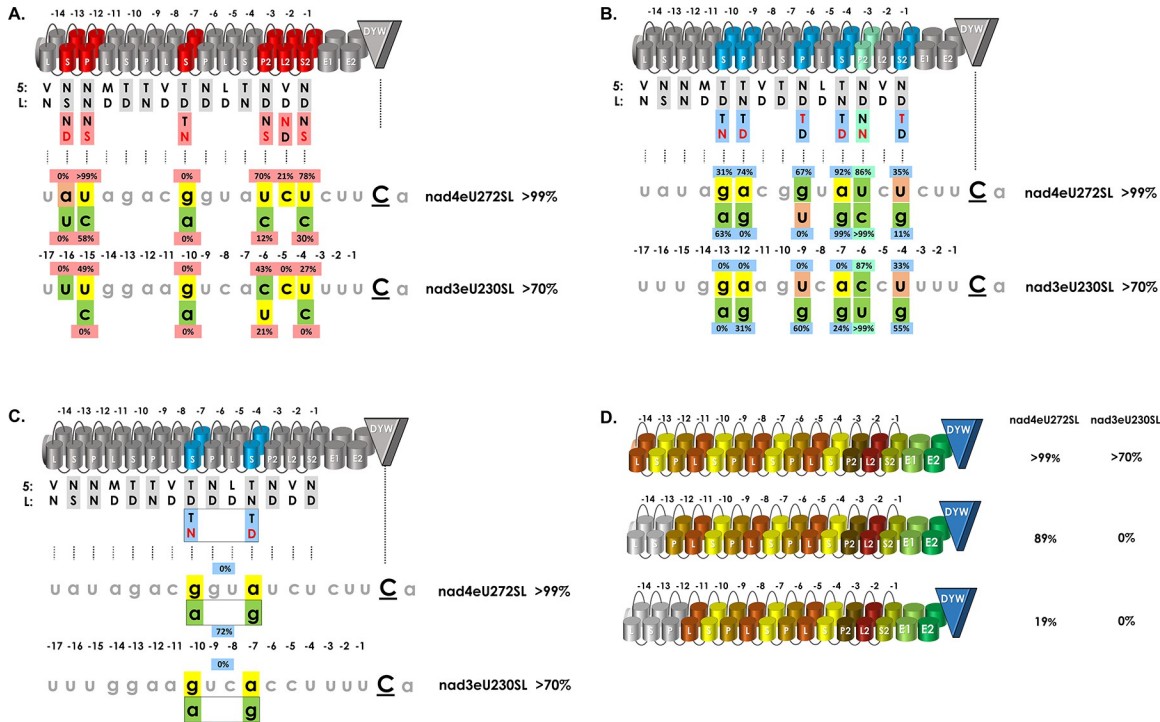

**Fig 3. Compensating and non-compensating PPR and target mutants.** Key positions '5' or 'Last' have been altered in individual PPRs of PPR56 (red font) in attempts of re-targeting to modified target sequences with conceptually improved matches in individual positions (green shading) of native targets nad4eU272SL and nad3eU230SL, respectively. RNA editing activities are indicated for the individual PPR mutants next to the respective target position identities. **A**. Target mutations not rescued by corresponding PPR mutations. No re-gain of RNA editing activity is observed for PPR mutations S-13NS>ND, P-12NN>NS, S-7TD>TN, P2-3ND>NS, L2-2VD>ND and S2-1ND>NS (red cylinders) juxtaposed with nucleotide positions -16, -15, -10, -6, -5 and -4 upstream of the edited cytidine in either target. **B**. Target mutations at least partially rescued by corresponding PPR mutations. Moderate re-gains of RNA editing activity are observed for at least one of the two targets for PPR mutations S-10TD>TN, P-9TN>TD, P-6ND>TD, S-4TN>TD and S2-1ND>TD (blue cylinders) opposite of nucleotide positions -13, -12, -9, -7 and -4, respectively. The green cylinder and shading indicates the mutated PPR P2-3ND>NN with a conceptually relaxed selectivity for U over C in position -6. **C**. Double target mutant. A double mutant PPR56|S-7TD>TN|S-4TN>TD shows no activity on the native targets but can be rescued to different amounts by the corresponding g-10a|a-7g target double mutants. **D**. N-terminal PPR truncations of PPR56. Progressive truncation of the two or three terminal PPRs of PPR56 lead to moderate or more drastic reduction of RNA editing efficiencies, respectively.

better match to cytidine by a ND>NS change did not improve editing of any target (Fig 3A). The changes introduced in the C-terminal P2-L2-S2 PPR triplet also included L2-2VD>ND leading to a drastic drop in RNA editing through this single amino exchange in an L-type PPR, which would be expected to have increased preference for pyrimidines in P- and S-type PPRs (Fig 3A). Most surprising, however, was the outcome of mutating the most N-terminal S-type PPR S-13NS>ND, which abolished RNA editing completely at both targets despite the mis-matching adenosine in that position in the *nad4* target. Introducing the conceptually fitting uridine in position -16 did not restore editing (Fig 3A).

Several other mutations in P- and S-type PPRs (S-10TD>TN, P-9TN>TD, P-6ND>TD, S-4TN>TD, P2-3ND>NN and S2-1ND>TD) had moderate consequences or could be rescued to a significant amount by corresponding changes in the targets (Fig 3B). The S-10TD>TN and the corresponding target mutant g-13a fits the general insight of an overall more resilient *nad4* target with reduced editing of the original target (31%) and higher editing of the adapted one (g-13a, 63%), while editing of the original *nad3* target and in the nad3eU230SL|g-13a mutant is abolished completely. The inverse mutation in the directly neighboring PPR P-9TN>TD again has only moderate effects on the *nad4* target (Fig 3B). However, and very

surprisingly, this mutant can only be rescued by the corresponding a-12g mutation in the *nad3* but not in the *nad4* target. Somewhat similar is the outcome for the PPR P-6ND>TD mutant.

Given the striking outcome of completely abolished RNA editing for the S-7TD>TN mutant that could not even be partially rescued by the corresponding g>a exchanges in the two targets (Fig 3A), we combined this mutation with the successful inverted exchange in S-4TN>TD (Fig 3B) in a double mutant (Fig 3C). Very surprisingly, this double mutant PPR56| S-7TD>TN|S-4TN>TD was able to edit both correspondingly adapted targets nad3eU230SL| g-10a|a-7g to 15% and nad4eU272SL|g-10a|a-7g to even 72%, indicating that the S-7TD>TN mutation does not cause a principally dysfunctional PPR56.

Overall, RNA editing factors characteristically show less conservation at the 5'-end of their PLS-type PPR arrays. However, the single amino acid mutation in PPR S-13NS>ND surprisingly abolished RNA editing and could not be rescued on the target side (Fig 3A). Effects were more moderate for mutating PPR P-12NN>NS. However, the original targets were still edited with higher efficiencies than the conceptually adapted ones with cytidines instead of uridines opposite to P-12NN>NS (Fig 3A). To further address this, we created two progressive N-terminal truncations of PPR56 (Fig 3D), either deleting PPR L-14 and the conceptually mismatching PPR S-13NS alone or a truncation including the following PPR P-12NN. For the shorter truncation RNA editing was abolished completely for the *nad3* target but only reduced to 89% for the generally more robust *nad4* target (Fig 3D). This result may be explained by the moderately better fit of S-13NS to the uridine in the *nad3* vs. the adenine in the *nad4* target. The further truncation including PPR P-12 further reduced RNA editing strongly at the *nad4* target (Fig 3D).

## The role of L-type PPRs

L-type PPRs only rarely feature amino acids in positions 5 and L that follow the PPR-RNA code rules. Notably, the two targets of PPR56 differ in the nucleotide identities opposite of its three central L-type PPRs L-11MD (a vs. g), L-8VD (c vs. a) and L-5LD (u vs. c). Hence, we mutated these positions to check whether they could contribute to the different RNA editing efficiencies observed for nad4eU272SL and nad3eU230SL (Fig 4). In a series of mutations adapting nucleotide identities to the respective other target, we find that changes in positions -14 (g<>a) and -8 (c<>u) do not significantly affect RNA editing in either target. Changes in position -11 (c<>a) decrease editing more significantly, however, and this is also the case after introducing a guanosine nucleotide in that position, eradicating editing for the *nad3* target altogether. Similar observations can be made for position -5 where the two native targets share a cytidine and the *nad3* target again proves to be more sensitive to changes. Notably, the corresponding triple-mutations converting positions -14, -11 and -8 to the identities in the respective other target decrease editing at the *nad4* target significantly to 26% and slightly improve editing at the *nad3* target to 76% (Fig 4). This, at first glance surprising, outcome may indicate that in some cases the concept of one-PPR-to-one-nucleotide matches may be too simplified and that certain successions of PPRs or target nucleotides could be disfavored for PPR-RNA interactions.

## The immediate environment of the editing sites

The general avoidance of a guanosine in position -1 immediately upstream of a cytidine to be edited has been recognized since long and is unequivocally supported by large editome data sets [50]. Moreover, there is increasing evidence that the E1, E2 and the DYW domains downstream of the PPR arrays can contribute to target recognition selectivity [49,51]. Accordingly,

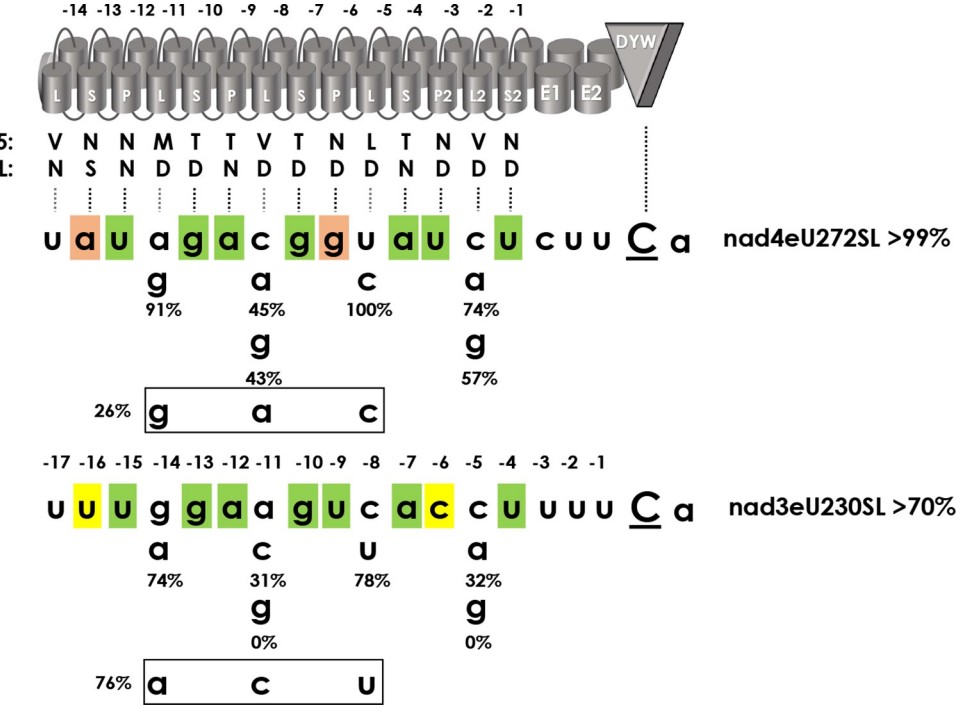

**Fig 4. Mutation of target positions opposite of L-type PPRs.** Target positions -14, -11 and -8 opposite of L-type PPRs L-11MD, L-8VD and L-5LD have been changed to the nucleotides present in the respective other native target of PPR56. Additional mutations to purines were introduced in positions -11 and -5 opposite of PPRs L-8VD and L2-2VD, which carry the same combination of amino acids in positions 5 and L and are mainly juxtaposed with cytidines in the targets. The strongest effects are seen for nad3eU230SL|a-11g and nad3eU230SL|c-5g abolishing RNA editing completely in the modified *nad3* targets. *Vice versa*, a much stronger effect is seen for the triple mutant nad4eU272SL|a-14g|c-11a|u-8c in the *nad4* target vs. the inverse changes nad3eU230SL|g-14a|a-11c|c-8u in the *nad3* target.

we also targeted positions in the immediate environment of the respective RNA editing sites for mutations (Fig 5). Exchanging the uridines in position -1 against guanosine indeed abolishes RNA editing altogether at both native targets of PPR56 (Fig 5). For other positions, the *nad3* target is again more affected, even by identical nucleotide exchanges in the same positions as in the *nad4* target. For example, this is clearly seen for target mutations both immediately downstream of the respective edits, i.e. nad4eU272SL|a+1u (>99%) vs. nad3eU230SL|a+1u (49%) and nad4eU272SL|u+2g (>99%) vs. nad3eU230SL|u+2g (61%) as well as upstream of the respective edits: nad4eU272SL|c-3u (>99%) vs. nad3eU230SL|u-3c (22%) or nad4eU272SL|u-2g (31%) vs. nad3eU230SL|u-2g (0%).

We tested for the possibility to artificially create stop or start codons through C-to-U editing, focusing on the *nad4* target that had proven to be significantly more tolerant against variations. Indeed, all three possible stop codons (UAA, UAG, UGA) could be efficiently created by editing after mutations in positions +1 and/or +2 with >99% editing efficiencies (Fig 5). Moreover, a combined nucleotide exchange in positions -1 and +1 (nad4eU272SL|u-1a|a+1g) also allows for artificial creation of a start codon by C-to-U editing quite efficiently (82%).

## RNA secondary structures inhibit, but native sequences further upstream enhance RNA editing

The binding of an RNA editing factor can certainly be expected to compete with RNA secondary structure formation by base pairing. Target point mutations were routinely tested for

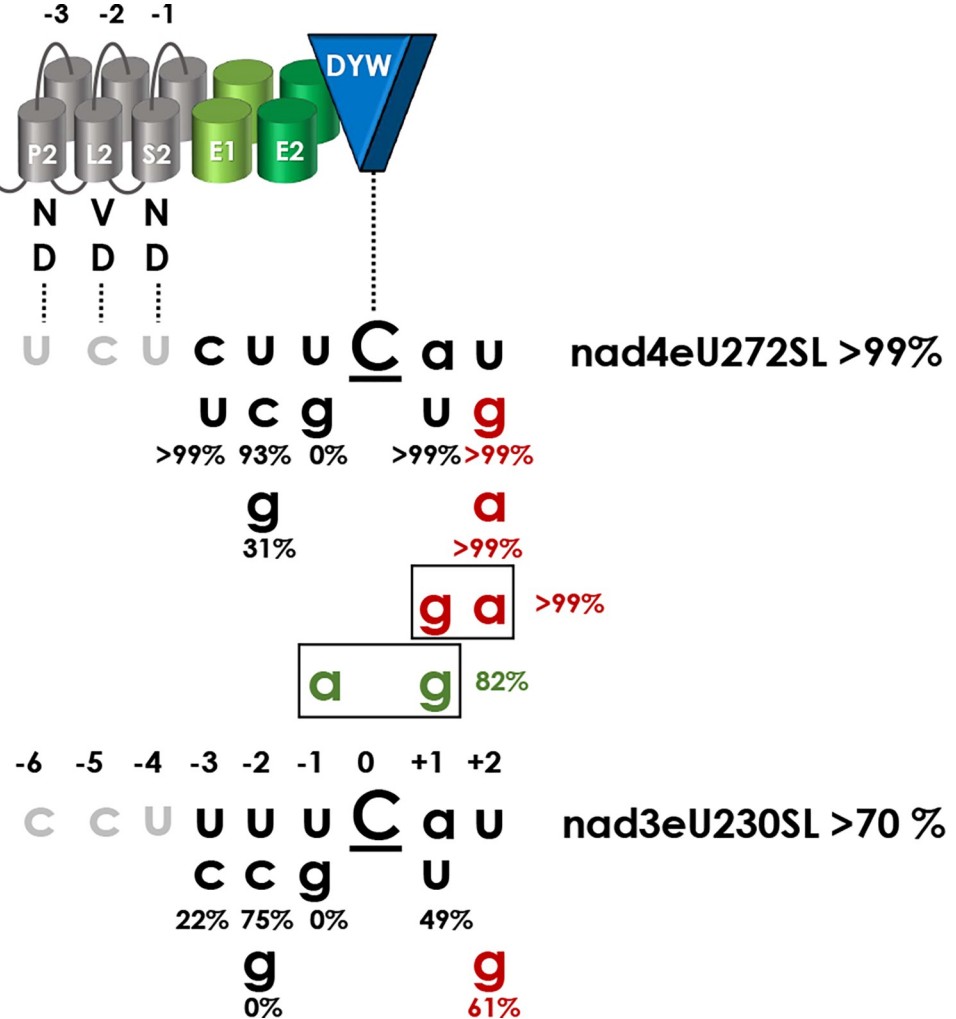

**Fig 5. Mutations around the RNA editing sites.** The two native targets of PPR56, nad4eU272SL and nad3eU230SL, feature identical nucleotides in positions -2 to +2 around the edited cytidines (uuCau). With the exception of the exchange u-1g eradicating RNA editing completely at both targets, other exchanges in the upstream region show different outcomes with nad4eU272SL|u-2g (31%) vs. nad3eU230SL|u-2g (0%) or the inverse pyrimidine exchanges in position -3 with no effect for *nad4* editing but reduction to 22% for *nad3*. Changes in positions +1 and +2 do not affect editing of the *nad4* target but reduce editing of *nad3*. The overall tolerance of the *nad4* target region against mutations in positions -1, +1 and +2 allows to engineer all three artificial stop codon identities (red) or an artificial start codon (green) to be created by C-to-U RNA editing.

potential secondary structure formations to exclude this as a potential cause for observed editing deficiencies [36]. We now intentionally created artificial secondary structures embedding the unchanged nad4eU272SL sequence targeted by PPR56 with upstream or with downstream sequences creating base-pairings with the core PPR target region (S3 Fig). An artificially added sequence upstream of the nad4eU272SL editing site potentially creating eight base pairs with positions -8 to -1 upstream of the cytidine editing left RNA editing efficiency unaffected whereas an extended region creating 13 base pairs reduced RNA editing activity to only 19% (S3 Fig). In contrast, RNA editing was abolished completely when artificial sequences were added behind position +5 relative to the cytidine editing target when creating potential base pairings with positions -10 to +1 or even only -8 to +1, respectively (S3 Fig).

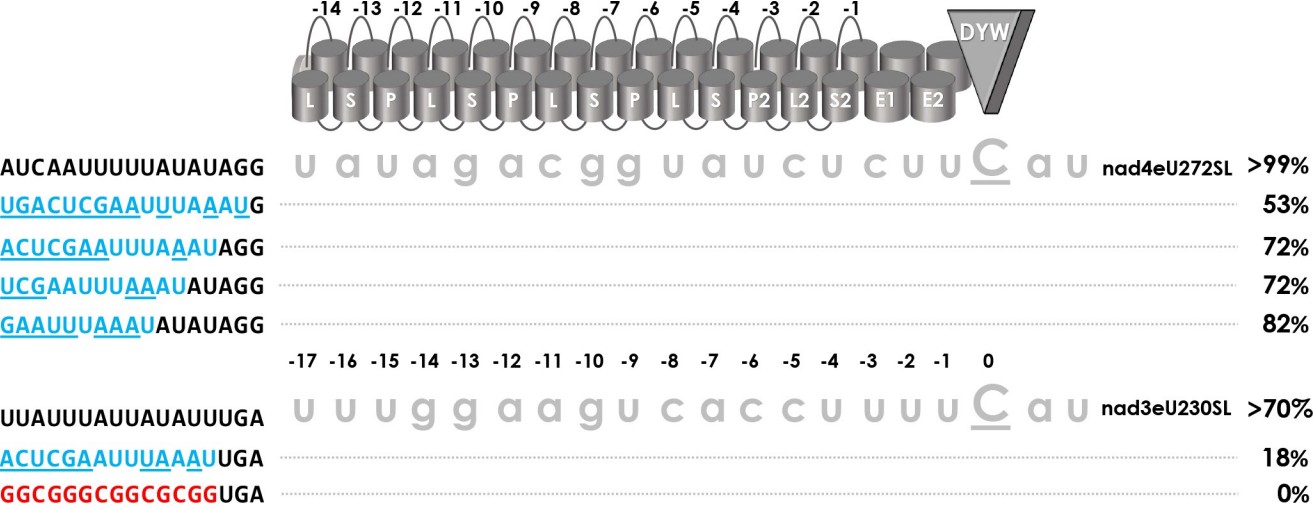

**Fig 6. The influence of sequences further upstream of targets.** PPR56 editing targets were cloned with 17 bp of additional native sequence upstream of the region supposed to be ultimately targeted by the PPR array, with the C-terminal PPR S2-1 juxtaposed with position -4 upstream of the editing site. Progressive 5'-truncations of this upstream sequence to only eight, seven, five or one nucleotide matching the native target behind the *SwaI* cloning site (AUUUAAAU) place them in closer proximity to the upstream vector sequences (blue) with nucleotides not matching the native upstream sequences underlined. The shortening results in serially decreased RNA editing activity to 53% for the *nad4* target. A yet stronger effect is seen for the *nad3* target where a 5'-truncation retaining four native upstream nucleotides reduces editing to 18%. Replacing the AU-rich sequence upstream of positions -20 with a GC-rich sequence (red font) abolishes editing at the nad3eU230SL site altogether.

Establishing the RNA editing setup in *E. coli*, the PPR56 targets were cloned to include 17 additional nucleotides of the native sequence further upstream of the sequence that is ultimately expected to be targeted by the PPR array [36]. We now tested whether these additional 5'- sequences had an effect on RNA editing efficiencies and found significant effects, indeed (Fig 6). Stepwise shortening the native target sequences at their 5'-ends progressively reduced RNA editing efficiencies considerably even though this would leave the expected core PPR-binding region of the target unaffected. Replacing the AU-rich region upstream of position -20 by a GC-rich sequence even abolished RNA editing at the nad3eU230SL target altogether (Fig 6). These results suggested that native sequences beyond the target ultimately bound by the PPR array may contribute to enrich PPR proteins in the neighborhood of the target or possibly even a 5'-to-3' sliding of the protein on the mRNA towards its ultimate binding position for C-to-U conversion.

## C-to-U RNA editing off-targets in the *E. coli* transcriptome

An initial screening of the *E. coli* transcriptome upon expression of PPR56 had identified 79 C-to-U RNA editing off-targets using strict criteria and confirmation from initially two independent RNA-seq replicates [36]. However, further candidates for C-to-U editing off-targets existed in the independent data sets that remained unconfirmed by the respective other replicate. We now created and analyzed four further RNA-seq data sets to screen for off-targets upon expression of PPR56 in constructs without or with different co-provided target combinations (S2 Data). Including the further replicates now resulted in the identification of altogether 133 off-targets (detected in a minimum of two independent data sets) for the wild-type PPR56 (Fig 7). The conservation profile for the 133 off-targets of wild-type PPR56 excellently confirms strong preferences for nucleotide positions opposite of P- and S-type PPRs as predicted from the PPR code in six cases: S-10TD:g, P-9TN:a, S-7TD:g, S-4TN:a, P2-3ND:u and S2-1ND:u. As generally known, we see a higher discrimination for the identities of purine than of

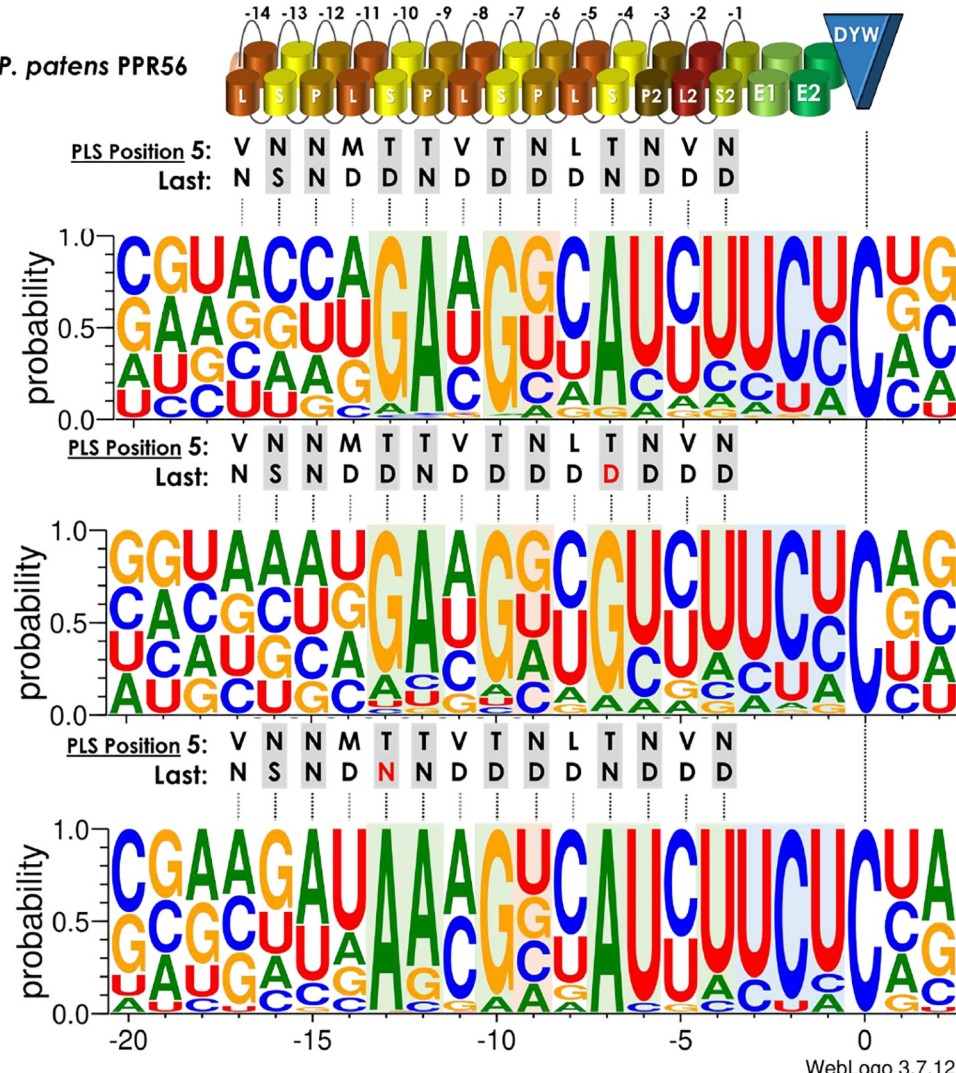

**Fig 7. Off-target analyses.** Off-targets of PPR56, PPR56|S4TN>TD and PPR56|S10TD>TN in the *E.coli* transcriptome summarized with Weblogo [89]. Consensus profiles were created from the sequences of 119, 382 and 15 C-to-U RNA editing off-targets, weighted with their respective editing efficiencies. Additional off-targets requiring nucleotide shifts for better binding matches (14, 67 and 1, respectively) were excluded for clarity (S2 Data). Modified positions in the PPRs are displayed in red. The mutated PPRs have a clear preference to the nucleotides fitting best to the modified binding amino acid pair in positions 5 and L according to the PPR-RNA code. Nucleotide preferences in positions −3, −2 and −1 are highlighted in blue. Nucleotide preferences within the PPR stretch and opposite to P- or S-motifs are highlighted in green.

pyrimidines. However, instead of an expected selectivity for uridine in position -9 opposite of PPR P-6ND we find a slightly stronger preference for guanidine. Notably, a guanosine is also unexpectedly present in the more efficiently edited native *nad4* target of PPR56. Additionally, there is strong selectivity for pyrimidines not only in positions -3 to -1 (mostly as UCU) but also in position -5 opposite of PPR L2-VD (Fig 7). Moreover, L-type PPR L-8VD appears to select against guanosine whereas no selectivity for pyrimidines is found in positions -16 and -15 opposite of PPRs S-13NS and P-12NN.

Additionally, we included RNA-seq analyses for three datasets each of the two PPR56 mutants with mutations in PPRs P-10TD>TN and S-4TN>TD, respectively (S2 Data). Intriguingly, the total number of off-targets is more than threefold (449 vs. 133) for the S-

4TN>TD mutant (Fig 7). This variant shows a strong shift in preference from adenosine to guanosine in position -7, exactly as expected from the PPR-RNA code. No further strong shifts of nucleotide preferences are observed for other positions in the conservation profile.

Mysteriously, exactly the opposite is observed for mutation of PPR56|S-10TD>TN where the number of off-targets is now drastically reduced from 133 to only 16. Expectedly, a strong selectivity for adenosine is now seen in position -13 juxtaposed with the mutated PPR as expected (Fig 7). Further judgements on potential other changes in the conservation profile also at other positions are not evident and should be considered with caution given the overall small number of only 16 off-targets in this case. It may be noted, however, that adenine or cytidine are prominently present here in position -11, corresponding to the identities in the two native targets opposite of PPR L-8VD, which had turned out to be most sensitive against changes (Fig 4).

## Serial combinations of PPR56 targets

The observation outlined above showing that native target sequences further upstream of the region juxtaposed with the PPR array contributed strongly for higher RNA editing activities (Fig 6) made us consider the possibility that multiplying targets on a single transcript may affect the respective RNA editing outcomes. The two known targets of PPR56 edited with high (*nad4*) and moderate (*nad3*) efficiencies offered an interesting test case allowing to check upon RNA editing activities at targets of PPR56 in varying combinations (Fig 8). Cloning the *nad3* target upstream of the *nad4* target led to a further reduction of nad3eU230SL RNA editing activity while leaving editing nad4eU272SL unaffected. A striking result was obtained, however, upon cloning the two targets in the reverse order (Fig 8). Again, nad4eU272SL editing remained unaffected but editing of nad3eU230SL site now rose to >99% indicating a beneficial effect of the upstream *nad4* target. This surprising enhancing effect of the upstream *nad4* target could even be seen more drastically for the previously tested *nad3* target variant where RNA editing was eradicated with a GC-rich sequence upstream of position -20 (Fig 6), where RNA editing activity is now boosted to 94% (Fig 8).

To check whether the enhancing effect of the upstream *nad4* target was dependent on its editability, we converted it into a "pre-edited" state replacing the target cytidine with thymidine (nad4eU272SL|c0u). Notably, the enhancing effect on the downstream *nad3* target remained unaffected, still resulting in >99% conversion at the nad3eU230SL target (Fig 8). However, introducing mutation nad4eU272SL|a-12g that creates a conceptual mismatch to PPR P-9TN and was found to abolish nad4eU272SL editing (Fig 2) into either the native or the pre-edited *nad4* target reduced the enhanced editing at the downstream *nad3* target to 93% or 86%, respectively (Fig 8).

We wished to check upon a potentially enhancing effect also on two selected off-targets of PPR56 in *E. coli* (S2 Data). Off- targets yegHeU419SL and folDeU-5 were edited to 38% and 78%, respectively, in the *E. coli* background transcriptome. However, only 38% of editing was observed for folDeU-5 and none at all for yegHeU419SL when cloned individually analogous to the native targets behind the PPR56 coding sequence. RNA editing of >99% or 17%, respectively, was observed when placed in tandem behind the upstream *nad4* target.

Surprisingly, however, we also observed an enhancing effect on RNA editing efficiency of the downstream *nad3* target to 93% when an artificial sequence introducing all possible transitions (a-g and c-u) into the upstream *nad4* sequence was used (Fig 8). Evidently, the sensitivity of editing the *nad3* target is not only reflected by changes in the PPR56 protein or in the target sequence but also by placement of the latter in the wider transcript environment as further confirmed below. We wondered whether the enhancing lateral effect on targets cloned in

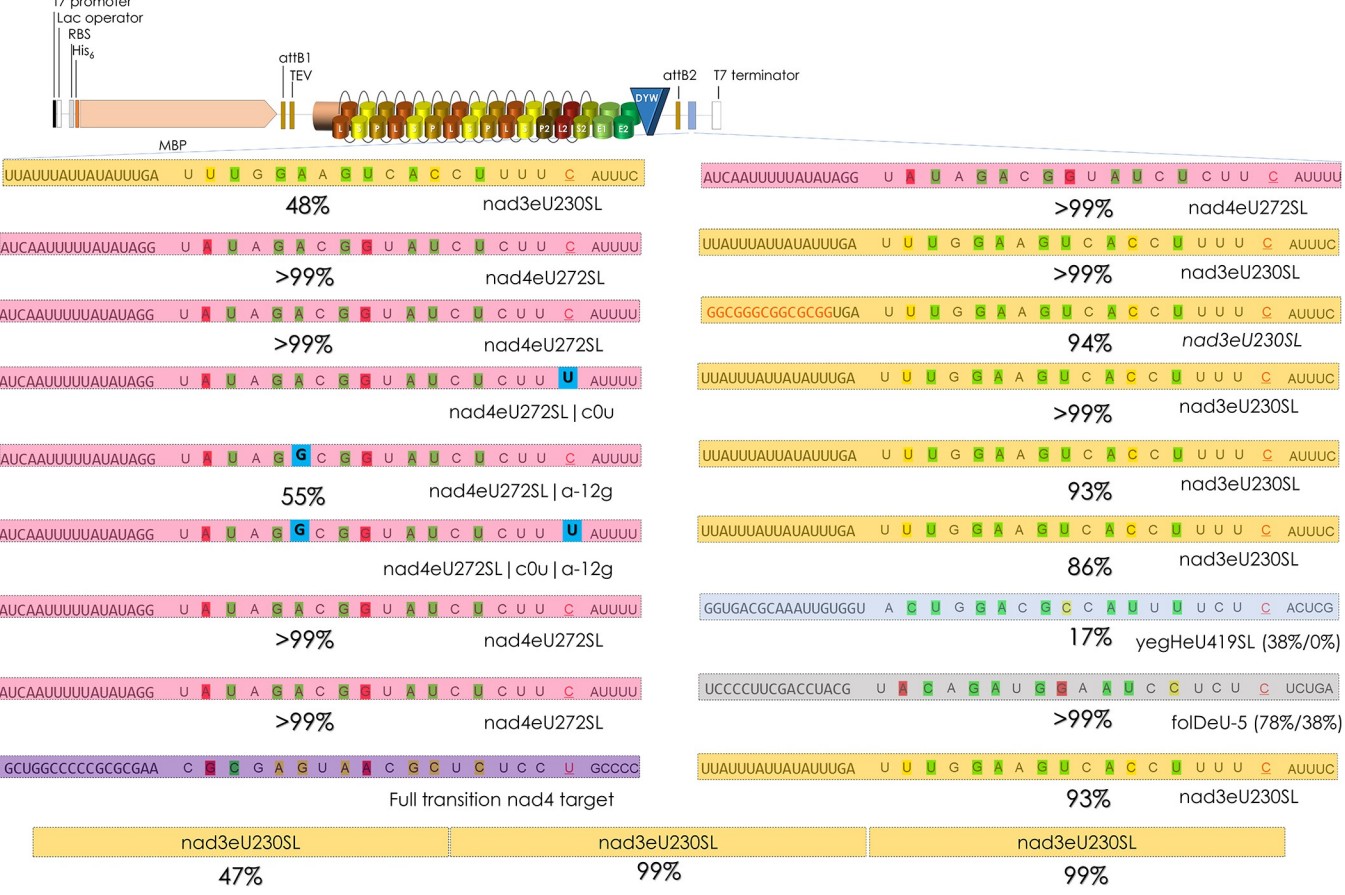

**Fig 8. Combining different PPR56 targets.** To test for mutual influences of combined targets on the same transcript, a series of tandem constructs and a triplicate arrangement of *nad3* targets was cloned in the multiple cloning site behind the PPR56 coding region. Shading highlights native targets nad3eU230SL (yellow) and nad4eU272SL (pink) and two off-targets identified in the *E. coli* background transcriptome in the transcripts of *yegH* (blue) and *folD* (grey). Numbers in parentheses indicate RNA editing efficiency observed in the off-target analysis and when cloned individually without the upstream *nad4* target, respectively. The series of constructs with the *nad4* upstream of the *nad3* target includes the one with the GC-rich sequence upstream of the latter (red font) that had abolished nad3eU230SL editing altogether.

tandem combinations could also be seen for the moderately efficient edited *nad3* target alone. Indeed, a triplicate arrangement of *nad3* targets resulted in diminished activity at the upstream-most copy, but enhanced RNA editing efficiencies at the middle and at the 3'-terminal target copy (Fig 8). Very much like the experimentation with truncation of the upstream extensions of the native targets (Fig 6) these findings indicate that sequences further upstream of the ultimate match of its PPR array to the RNA editing target can significantly affect activity of an RNA editing factor.

## Placement of targets towards the 5' or 3'-end of a long RNA

We wished to test placement of targets in different positions and made use of the newly constructed vector pET41Kmod2 (S2 Fig), which allows the alternative cloning of targets also upstream of the editing factor coding sequence into the 5'-UTR. A combination of the *nad4* target in the 5'-UTR with the *nad3* target in the 3'-UTR could not enhance editing of the latter while the former remained unaffected (Fig 9A). Surprisingly though, cloning in the inverse arrangement led to significant increase in editing at the nad3eU230SL target when cloned into the 5'-UTR (Fig 9A). This held equally true for tandem cloning of the two targets into the 5'-

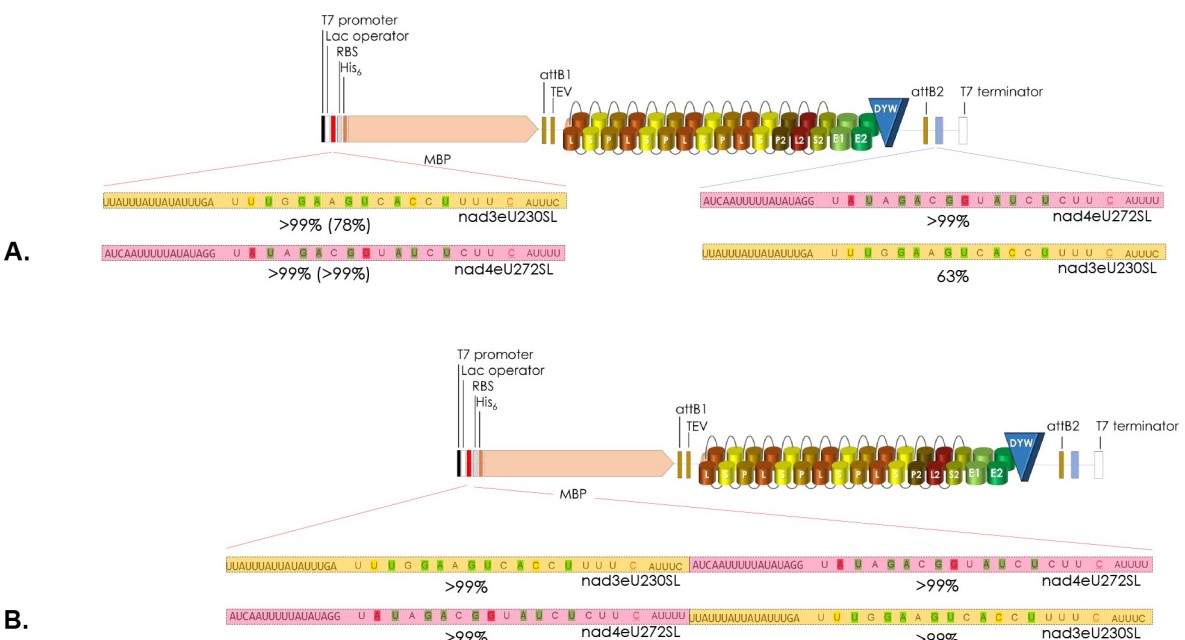

**Fig 9. RNA editing target placement at the 5'- or 3'-end of a long mRNA.** A. The two native targets of PPR56 were placed separately into the previously used 3'-MCS downstream of the protein coding sequence (blue lines) and into the newly created 5'-MCS (red lines) in pET41Kmod2 (S2 Fig) in both alternative combinations. Editing efficiencies for single targets inserted in 5' MCS are given in brackets. Cloning is done via *Not*I-*Pac*I in the 5'-MCS and via *Swa*I-*Asc*I in the 3'-MCS. **B**. The tandem combination of the two targets previously tested in the 3'-MCS was now also tested in the 5'-MCS.

UTR in either orientation (Fig 9B). Evidently, providing the "weak" *nad3* target in a 5'- rather than in a 3'-UTR appears to allow for better access and more efficient editing, aside from the enhancing effect of tandem target arrangements.

Resulting from the above findings, we tested five additional off-targets identified in *E. coli* (fdhEeU403Q*, paoCeU542TM, rarAeU407TI, arnAeU242SF and cydCeU980PL) that showed variable editing efficiencies at different RNA read coverages and different matches to the PPR array of PPR56 (Fig 10A). Towards that end we tested both for an effect of tandem-cloning with the upstream *nad4* target (Fig 10B) as a possible enhancer as well as for their placement in the 5'-MCS in wide distance from the downstream *nad4* target (Fig 10C). In three cases we found that RNA editing could be strongly enhanced both by placing the respective off-target either in tandem behind the native *nad4* target or alternatively into the 5'-MCS distant from the nad4eU272SL target located in the 3'-MCS: rarAeU407TI from 24% to 66% or 70%, fdheU403Q* from 16% to 75% or 61% and for cydCeU980PL from 50% to over 99% with both placements, respectively. However, a striking reduction was found to only 4% for arnAeU242SF with both cloning strategies and even to the abolishment of editing for pao-CeU542TM in the tandem cloning approach (Fig 10B). Notably, in the latter case RNA editing at the native nad4eU272SL site was concomitantly also reduced to 62% while the usual highly efficient editing was observed in the other nine constructs.

## Exploring novel candidate targets

It is important to keep in mind that orthologues of a functionally characterized plant RNA editing factor may have additional or different functions in other species. Intriguingly, the two targets of PPR56 in *Physcomitrium patens* are not conserved in most other available moss mitogenomes (with exceptions in the Pottiaceae), but rather exist in a pre-edited state with

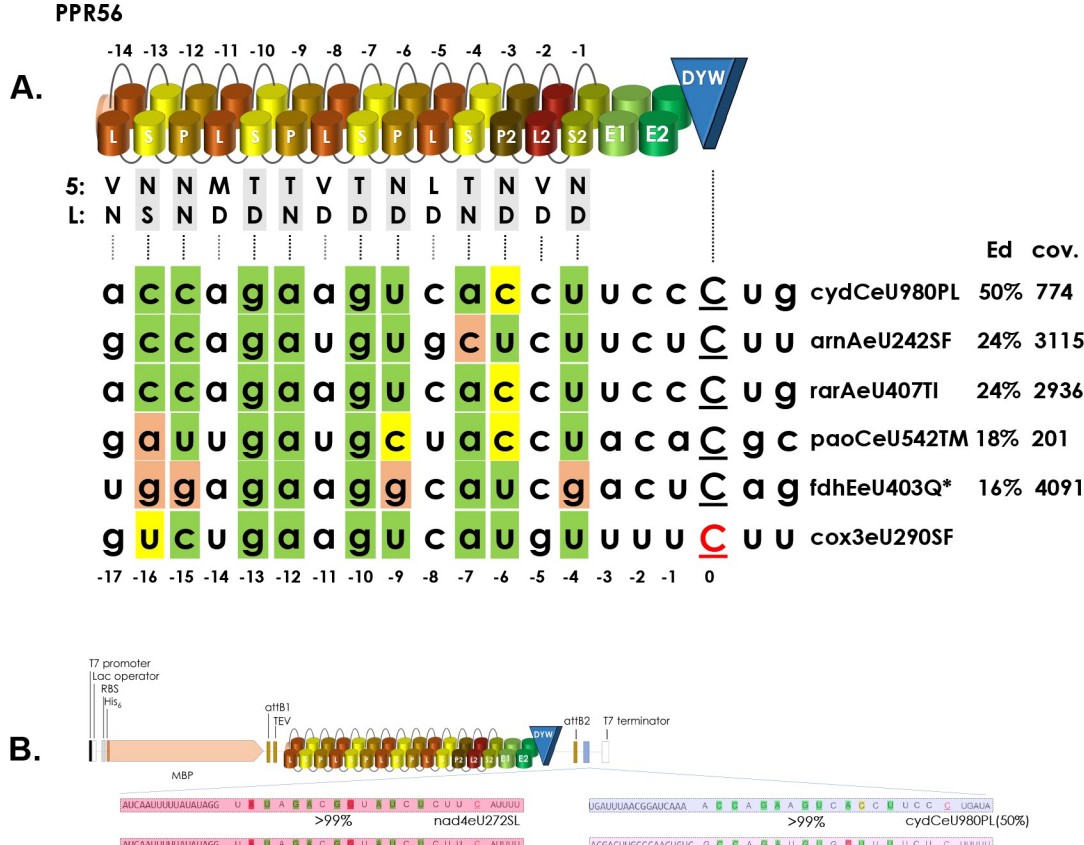

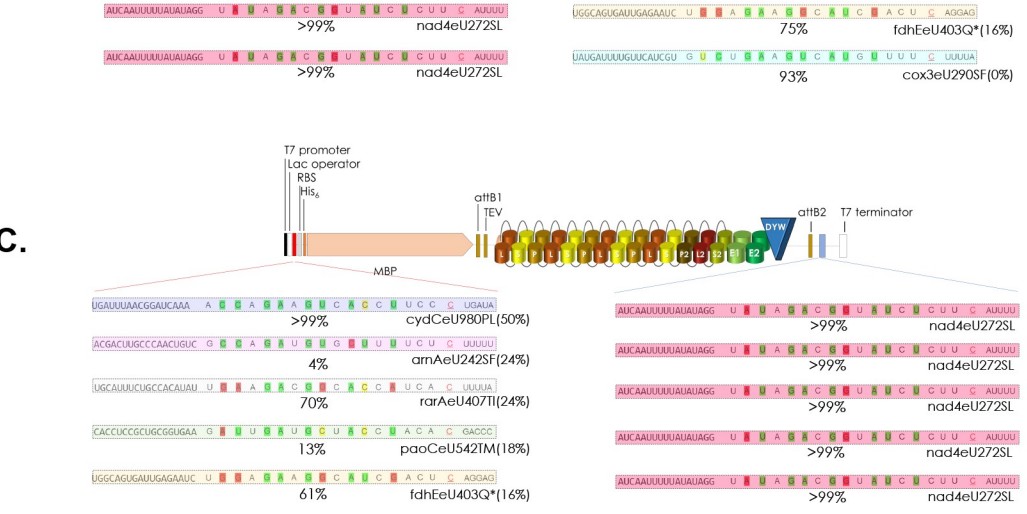

**Fig 10. Off-targets in different cloning positions.** Five off-targets of PPR56 identified in *E. coli* characterized by different RNA coverages and editing efficiencies (A) were selected for cloning in tandem behind the native nad4eU272SL target of PPR56 (B) or separately into the upstream MCS in the 5'-UTR (C). Editing efficiencies for off-targets in E. coli transcripts are given in brackets. Enhancement of RNA editing was found for three of the off-targets (cydCeU980PL, rarAeU407TI and fdhEeU403Q*) in either cloning arrangement and also for the, hitherto hypothetical, candidate editing cox3eU290SF when cloned downstream of nad4eU272SL (B).

thymidines in the genomic positions. Using the TargetScan option of PREPACT [50] we wished to find alternative targets for PPR56 that may exist in a pre-edited state with thymidine present in the mitogenome of *Physcomitrium*. Indeed we could find cox3eU290SF as such as potential target matching excellently to the RNA binding properties of PPR56 (Fig 10A). The *E. coli* RNA editing assay setup allows to test such a hypothesis quickly and we accordingly exchanged the T at the potential editing position of the *Physcomitrium* mtDNA sequence into a C. Whereas we could not detect editing of cox3eU290SF when routinely cloned as a single target inserted downstream of the PPR protein coding region, we observed an editing efficiency of 93% when cloned in tandem downstream of nad4eU272SL (Fig 10B). At present, cox3eU290SF cannot be identified as a candidate editing site in moss mtDNAs but is confirmed as an RNA editing site in the mitochondria of the lycophytes *Isoetes engelmannii* [52] and *Selaginella moellendorffii* [53] and in the fern *Haplopteris ensiformis* [54].

## Discussion

### Mutating the DYW domain: different effects on two native targets

All of our experimentation showed that the *nad4* target of PPR56 is more resilient towards changes both on the target side and on the protein side than the *nad3* editing target site, which proved to be much more sensitive. Notably, the higher sensitivity of the *nad3* target towards changes also extended to mutations in the DYW domain of PPR56 (Fig 1B). The carboxy-terminal DYW domain of plant RNA editing factors has long been suspected, and is meanwhile well confirmed, as the catalytic cytidine deaminase domain [36,46,48,55–57]. Many of the highly conserved amino acid residues in the DYW domain are essential for functionality as here again confirmed with a set of mutations in the DYW domain of PPR56. However, while six mutants with single amino acid exchanges in the DYW domain of PPR56 lost RNA editing activity on both targets, seven others affected RNA editing at the nad3eU230SL target more strongly than at the nad4eU272SL site (Fig 1B). This is all the more striking given that target positions -2 to +2 around the cytidine targeted for C-to-U conversion are identical for the two targets of PPR56. Evidently, the DYW domain is not simply a flexible enzymatic unit that can easily be transplanted but relies on intricate interactions of the upstream protein regions with different RNA targets. Notably, the *nad4* target of PPR56 not only tolerates exchanges in positions +1 and +2 allowing for the artificial creation of stop codons through C-to-U RNA editing but also for the artificial creation of a start codon after conversion of position -1 to adenosine (Fig 5).

### PPR arrays: The P- and S-type PPRs

It is generally understood that the upstream PPR array of a plant RNA editing factor is responsible for proper target recognition following the established PPR-RNA code rules [41–44,58]. PPR56 is no exception but it should be noted that its P- and S-type PPRs show overall even a slightly better fit to its more weakly edited target nad3eU230SL than to its strongly edited target nad4eU272SL (Fig 1A). Target selectivity following the PPR code is excellently reflected by the off-target conservation profiles fitting expectations for three P-type and three S-type PPRs of PPR56, including an intended re-targeting after changing key positions in two of these PPRs (Fig 7). However, exceptions exist as seen for P-type PPR P-6ND which unexpectedly appears to select for guanidines as well as for uridines (Fig 7), possibly as part of the explanation for efficient editing of nad4eU272SL with a guanidine in the corresponding target position -9. However, RNA editing is lost at the u-9g target mutant of nad3eU230SL (Fig 2) and this is just one of several examples found in the course of our work showing restricted predictability for RNA editing activities even upon small molecular changes.

Another dramatic example is a single u-to-c transition in position -15 of the targets which leaves the high editing efficiency at nad4eU272SL unaffected but abolishes editing completely for the nad3eU230SL target (Fig 2). This is quite surprising given that the N-terminal PPRs generally contribute more weakly to target selectivity and, fitting this general assumption, the off-target conservation profiles show no strong preference in these positions (Fig 7).

Similarly, the behavior of PPR56 protein variants is predictable only to a limited degree. For mutations in the crucial positions 5 or L of P- and S-type PPRs of PPR56 we found that ca. 50% of them could be rescued to variable degrees by corresponding mutations on the target side for at least one of the native targets (Fig 3B). However, this was not the case for the other 50% of mutants tested (Fig 3A). The PPR protein mutants with successful retargeting included S-10TD>TN and S-4TN>TD that were also tested for off-targets in *E. coli*. Intriguingly, PPR mutant S-4TN>TD not only proved to be more resilient on the *nad4* target and to be rescued by a>g exchanges in native targets (Fig 3B), but also resulted in a more than threefold amount of 449 off-targets compared to 133 in wild-type PPR56 (Fig 7). Exactly the opposite is observed for PPR mutant S-10TD>TN having a stronger impact that cannot be rescued on the *nad3* target and resulting in a strictly reduced set of only 16 off-targets (Fig 7). A similar, although not quite as drastic effect has recently been found for another PPR re-targeting mutant S-7TD>TN in human cells while a huge increase in off-targets was also seen for the S-4 TN>TD mutant [37]. We conclude that the observed effects are very unlikely an effect of the bacterial vs. the eukaryotic expression setups but rather inherent to the PPR array and strongly point to significant impacts on overall protein features even upon changes of single amino acids in a dedicated PPR. Individual PPRs appear to contribute very differently to target recognition or ultimate RNA editing efficiencies and even single amino acid exchanges in position 5 or L of a PPR may strongly increase or decrease the flexibility of an RNA editing factor for target recognition. In this context it should be remembered that several point mutation alleles also outside of positions 5 or L in PPRs of functionally characterized RNA editing factors strongly affected specific RNA editing functionality in yet unclear ways, e.g. [59]. A G-to-R mutation in the DEK45 protein is another recently reported example along those lines [60].

## PPR arrays: The L-type PPRs

The contribution of L-type PPRs for target recognition has been investigated previously, ascribing them a role in RNA editing but not in RNA binding [61]. Notably, the two native targets of PPR56 display different nucleotides opposite of their three central L-type PPRs (Fig 4). Creating target mutants replacing the nucleotides with the respective other showed clear effects only for PPR L-8VD (Fig 4). Nevertheless, the cytidine-to-adenosine exchange in the *nad4* target as well as the inverse exchange in the *nad3* target position -11 juxtaposed with PPR L-8VD both reduced RNA editing efficiency (Fig 4). Remarkably, however, the reduced off-target data set for the PPR mutant S-10TD>TN in particular shows a clear preference for adenosine or cytosine in this position, matching the nucleotide identities in the two native targets (Fig 7).

## RNA editing efficiencies and the wider transcript context

Using target predictions based on the PPR-RNA code generally finds many additional candidate RNA editing sites with equal or even better matches than the documented targets of an RNA editing factor, but these sites remain unedited. To some extent, RNA secondary structures may play a role to explain this observation. Placing the cytidine to be edited in the context of RNA secondary structures can reduce or even abolish RNA editing altogether (S3 Fig). In case of the two closely spaced mitochondrial editing sites ccmFCeU103PS and ccmFCeU122SF

in *P. patens*, the upstream located editing site needs to be addressed by PPR65 first, most likely to destabilize a secondary structure to allow PPR71 to bind and edit the downstream site [62]. Such observations can certainly be expected given that binding of a PPR protein to RNA must compete with RNA secondary structure formation. This has been investigated systematically previously, e.g. for the P-type protein PPR10 [63]. Particularly interesting will be the further functional characterization of RNA editing factors like DEK46 acting on edited cytidines naturally embedded in stable secondary structures such as domain V of group II introns [3,35,64]. However, reliable prognoses on a RNA secondary structures are mostly limited to small transcripts while predictions of long-range base-pair formations *in vivo* is questionable.

Maybe more importantly, we here found that several transcript features beyond the region ultimately targeted by the PLS-type PPR array strongly contribute to attract and/or enhance the activity of an editing factor like PPR56. With the benefit of hindsight it has likely been helpful that 5'-extensions beyond the core PPR-targeted region have been included initially in the establishment of the heterologous editing systems [36,37]. We now found that additional sequences upstream of the RNA sequence ultimately targeted by the PPR array have a significant influence on efficient RNA editing. Progressive 5'-deletions of the native targets and their replacement with foreign sequences results in stark reduction of RNA editing up to complete loss in the case of the "weak" *nad3* target despite retention of native sequence 20 nucleotides upstream of the cytidine to be edited.

*Vice versa*, we find that within tandem arrangements, an upstream sequence is able to enhance RNA editing at downstream targets and this is independent of a cytidine present for conversion to uridine in the upstream "enhancer" sequence. Notably, it may be interesting to remember that an enhancing effect of multiplied targets had also been observed in early *in vitro* experimentation [65]. With the enhanced system, we were also able to identify cox3eU290SF as a new additional target in the mitochondrial transcriptome of *P. patens*, which can be edited, when a C is introduced at the editing position.

Designing our setups for heterologous expression, we placed the editing targets into the 3'-UTR behind the editing factor coding sequences, which was intended to test for RNA editing by subsequent cDNA analysis restricted to full length mRNAs. Surprisingly, we now find that not only tandem target arrangements but also their alternative placement into the 5'-UTR can enhance RNA editing to >99% (Figs 8–10).

## Conclusions and outlook

It is likely unsurprising that heterologous functional expression in prokaryotic and eukaryotic setups and for *in vitro* studies succeeded with evolutionary ancestral RNA editing factors comprising all necessary functionalities in just one polypeptide [36,37,46,48,66,67]. All available data for PPR56 show very similar behavior upon heterologous expression in the bacterial or human cells and even despite differently fused protein tags, indicating its independence from prokaryotic or eukaryotic host factors or from the many other plant organelle RNA maturation factors [68]. Functional heterologous expression will be much more complex for multiprotein editosomes that have to assemble for RNA editing in flowering plants to reconstitute target recognition and a DYW-type cytidine deaminase or to enhance RNA-binding capacities with MORFs/RIPs by protein-protein interactions [69–72].

PPR proteins are frequently investigated by *in vitro* experimentation with REMSAs (RNA electromobility shift assays) using RNA oligonucleotides representing the region bound by the PPR array. Such experimentation has contributed tremendously to understand their mode of binding and may be entirely sufficient for the study of P-type PPR proteins, which largely stabilize transcript ends by tight binding to an RNA, for example. However, scenarios may differ

for the PLS-type PPR proteins like RNA editing factors, which are expected to bind only temporarily to allow for cytidine deamination. The *in vivo* experimentation in *E. coli* reported here strongly suggests that the wider transcript environments and the placements of targets matter significantly for the ultimately detected RNA editing frequencies.

We here report that several circumstances affect RNA editing efficiencies even for "simple" single-polypeptide RNA editing factors like PPR56, including (i) the enigmatic L-type PPRs, (ii) the RNA sequences further upstream of the region ultimately bound by the PPR array, (iii) the tandem combination of targets or (iv) their respective placement in long transcripts as here exemplarily shown for the 5'- and 3'-UTRs flanking the PPR56 coding region with our modified vector setup. Whether binding preferences of individual PPRs in plant editing factors can be simply changed via modification of the 5th or last amino acid appears to very much rely on their respective position and/or the overall structure of the PPR array. Hence, any future experimentation with native RNA editing factors or those based on artificial "designer" PPR arrays [66,67,73–80] should take the above into account for testing and conclusions.

## Materials and methods

### Molecular cloning

Cloning for expression of *Physcomitrium patens* PPR56 variants and targets in *Escherichia coli* was based on vector pET41Kmod as outlined earlier [36]. PPR56 coding sequences (lacking the N-terminus with the signal peptide and including only 14 amino acids upstream of the first clearly identified PPR) are cloned via gateway cloning downstream of an N-terminal $His_6$ tag and the maltose-binding protein (MBP) for improved protein solubility [81] behind a T7 promoter controlled by the lac operator. RNA editing target sequences were cloned behind the protein sequence upstream of a T7 terminator. Here, we also created a new vector variant pET41Kmod2 (S2 Fig) with further restriction sites allowing for cloning targets also upstream of the respective coding region. To that end, we made use of a former *XbaI* site to create a *NotI-EcoRI-PacI-PstI* multiple cloning site (MCS) upstream of the ribosome binding site (RBS) in pET41Kmod. Target sequences including flanking restriction sites were generated with synthesized oligonucleotides for both DNA strands (Integrated DNA technologies Europe, BVBA, Leuven, Belgium) and ligated into dephosphorylated vectors after hybridization and phosphorylation. All oligonucleotides used in the course of this work are listed in S3 Data. To introduce site-directed mutations into PPR56 coding sequence we used an overlap PCR strategy with mutagenizing oligonucleotides. N-terminally truncated PPR56 coding sequences were amplified with classic PCR approaches using Phusion High-Fidelity DNA Polymerase (Thermo Fisher Scientific) as described [36] to retain 14 native amino acids upstream of the most N-terminal completely retained PPR (Fig 3D)

### Protein expression and analysis of RNA editing

The setup for the expression of different constructs in the heterologous *E. coli* system and the downstream analysis of RNA editing was done as outlined previously [36]. Briefly, 25 mL of *E. coli* Rosetta 2 (DE3) cultures were pre-grown in 100 mL Erlenmeyer flasks with baffles in LB medium supplemented with 50 μM kanamycin, 17 μM chloramphenicol and 0.4 mM $ZnSO_4$ at 37˚C until reaching an $OD_{600}$ of ca. 0.5. The bacterial cultures were then cooled on ice for 5 min. before adding 0.4 mM IPTG for induction of expression and incubation for 20 h at 16˚C and 180 rpm. To further explore the expression system, we here also tested elevated incubation temperatures of 24˚C instead of the routinely used 16˚C for incubation after induction of expression (S4A Fig) and shorter incubation times of only 4 h or 8 h, respectively, instead of the routinely used 20 h incubation time before harvest and analysis of RNA editing (S4B Fig).

These experiments suggested to further use a 20 h incubation time at 16°C routinely, although shortened incubation times may be warranted to differentiate between constructs when very high RNA editing activities are observed. RT-PCR sequencing chromatograms were analyzed with MEGA 7 [82] and Bioedit 7.0.5.3 [83]. RNA editing was quantified by the ratio of the thymidine peak to the sum of thymidine and cytidine peaks in the editing position. RNA editing was routinely checked for three biological replicates, i.e. three independent bacterial clones after re-transformation of a given plasmid construct after control sequencing. PPR56 protein variants were routinely checked for expression on SDS-PAGE gels. Mutant proteins not revealing RNA editing were additionally checked by solubility tests as outlined previously [46] using monoclonal antibodies against His$_6$ (His.H8, Invitrogen) and secondary antibody Rabbit anti-Mouse IgG (H+L) (Invitrogen).

## Total RNA sequencing and off-target detection

To identify off-targets in the *E. coli* transcriptome, total RNA was prepared from individual experiments by using the Nucleo-Spin RNA kit (Macherey-Nagel), followed by DNase I treatment (Thermo Fisher Scientific). Library preparation was done after rRNA depletion (TruSeq Stranded Total RNA with Ribo-Zero), followed by Illumina sequencing (150 bp paired-end with NovaSeq 6000) done by either Novogene or Macrogen. To generate construct-specific DNA reference reads, the simulated reads (by ART MountRainier version 2016-06-05) of pET41Kmod with PPR56 and respective target sequences were merged with genomic DNA reads (WTDNA_SRR941832) of BL21(DE3) cells [84]. The construct-specific reference was made by merging pRARE2 sequence (Rosetta Competent Cells, 70953; Millipore, San Diego, CA), pET41Kmod with respective constructs and the *E. coli* BL21 genome (CP010816.1). The datasets obtained are summarized in S2 Data. After quantifying the RNA-seq raw data by FastQC (https://www.bioinformatics.babraham.ac.uk/ projects/ fastqc/), the transcriptome reads were aligned with construct-specific DNA reads against the construct-specific reference by GSNAP v2020/04/08 [85] with proposed settings [86]. The SNPs were called by JACUSA v1.3 [87]. The SNPs were further restricted by a custom-made R script (established with kind help provided by S. Zumkeller) restricting to SNPs obtained in at least two datasets from expression of the same protein but not in wild-type or expressing other editing factors like PPR65 [36]. Final RNA editing efficiency was calculated by adding up total RNA reads from all hitting datasets at a site. RNA editing sites were only considered for sites with (i) RNA read coverage of at least 30, (ii) a clear signal for transition in the RNA reads (T+C or G+A > 99%), (iii) a clear DNA reference position (G or C > 98%) and (iv) a C-to-U RNA signal of at least 1%. The original SNP mapping data are given in S2 Data. Primary data have been deposited at the NCBI under BioProject accession number PRJNA984633.

## Supporting information

**S1 Fig. WebLogo conservation profile of the DYW domains in nine *Physcomitrium patens* RNA editing factors.** The conservation plot based on the alignment of the DYW domains of nine functionally characterized RNA editing factors of *Physcomitrium patens* has been obtained with WebLogo [89]. Highlighted with frames are the characteristic PG box at the N-terminus of the DYW domain, the signature motifs for coordination of two zinc ions including the catalytic center (HSE) of the cytidine deaminase and the region of amino acids 37–42 discussed as relevant for compatibility for creating protein chimeras [49]. The "gating domain" as recently defined from X-ray structural analysis after crystallization of the OTP86 DYW domain [46] is highlighted in orange. Several residues have been selected for the study of

mutants (Fig 1B).
(TIF)

**S2 Fig. Expression vector system pet41Kmod2.** Vector pET41Kmod for expression of RNA editing factors and their targets has been reported previously [36]. Coding sequences of RNA editing factors are inserted by Gateway cloning resulting in flanking attachment attB sequences connecting in-frame via a TEV cleavage site to the upstream maltose binding protein (MBP) and an N-terminal His$_6$ tag. Transcription is driven from a T7 promoter controlled by a lac operator and translation is initiated by a ribosome binding site (RBS). PPR56 is cloned with an N-terminal extension of 14 native amino acids upstream from its N-terminal PPR L-14. Target sequences were designed with hybridized oligonucleotides inserted by classic cloning into a multiple cloning site (MCS, *Swa*I-*Hin*dIII-*Asc*I-*Bst*BI) in the 3'-UTR between attB2 and a T7 terminator. A new vector variant pET41Kmod2 has been created which also allows for cloning target sequences alternatively upstream into the 5'-UTR in a second MCS (*Not*I-*Eco*RI-*Pac*I-*Pst*I) inserted into a previous *Xba*I site. The vector map was created with SnapGene Viewer 6.2.1 (https://www.snapgene.com).
(TIF)

**S3 Fig. The influence of RNA secondary structures embedding the editing site.** Artificial sequences have been added upstream (yellow) or downstream (green) to embed the cytidine targeted for RNA editing (red) into secondary structures. The sequence upstream of the cytidine editing target that is supposedly juxtaposed with the PPR array of PPR56 (see Fig 1A) is shown in small letters. The RNAfold WebServer of the ViennaRNA package [90] was used to predict the secondary structures. RNA structure models were created with VARNAv3-93 (https://varna.lri.fr).
(TIF)

**S4 Fig. Temperature- and time-dependence of RNA editing.** A. RNA editing was checked at an elevated temperature of 24° (orange bars) instead of the routinely used 16°C (blue bars) for heterologous protein expression in the *E. coli* Rosetta 2 (DE3) arctic express system for a selection of altogether twelve constructs. The elevated temperature of 24°C generally disfavors RNA editing compared to incubation at 16° both on *nad4* and on *nad3* targets with the interesting exception of the PPR56|DYW:P2A mutant. B. RNA editing was checked for eight selected constructs also at shorter incubation times of only 4 h or 8 h, respectively, instead of the routinely used 20 h of incubation at 16°C after induction of expression. A reduction of RNA editing is seen in all cases of shorter incubation times except for the efficiently edited *nad4* target, which already shows >99% editing after 8 h of incubation.
(TIF)

**S1 Data. Full set of *E. coli* RNA editing assays.** Full table of results for all individual *E. coli* RNA editing assays including standard deviations. C-to-U RNA editing frequencies are given as 100% when no remaining cytidine signal was detectable upon sequencing of RT-PCR products.
(XLSX)

**S2 Data. RNA-seq data sets for analysis of off-targets in *Escherichia coli*.** RNA-seq datasets analyzed for C-to-U RNA editing off-targets. Separate tabs for the summary off-target lists for PPR56, PPR56|S-4TN>TD and PPR56|S-10TD>TN and 13 individual data sets for JACUSA variant calls (*E. coli* wild-type background control for reference, native PPR56 without co-delivered targets (2 replicates), with co-delivered nad3eU230SL target, nad4eU272SL target (2 replicates) and combined *nad4-nad3* target, PPR56|S-10TD>TN without or with co-delivered

target nad4eU272SL or nad4eU272SL|g-13a, and PPR56|S-4TN>TD without or with co-delivered target nad4eU272SL or nad4eU272SL|a-7g) analyzed in the course of this study.
(XLSX)

**S3 Data. Oligonucleotides.** Oligonucleotides used in this study. All oligonucleotides were synthesized by IDT (Integrated DNA technologies Europe, BVBA, Leuven, Belgium).
(XLSX)

## Acknowledgments

We gratefully acknowledge the computer resources and support provided by the Paderborn Center for Parallel Computing (PC²). We wish to thank Bastian Oldenkott, Philipp Gerke and Simon Zumkeller in our group for the establishment and help in further development of bioinformatic pipelines and Sarah Brenner for technical assistance. We especially like to thank Elena Lesch for establishing the program to generate construct-specific DNA references. We also like to thank Bastian Oldenkott for designing the initial PPR model for PPR protein figures. We thank Mark Hermann Vegas and Grazia Margherita Willerscheidt for cloning constructs and performing initial *E. coli* experiments as part of their experimental Bachelor theses work.

## Author Contributions

**Conceptualization:** Yingying Yang, Mareike Schallenberg-Rüdinger, Volker Knoop.

**Data curation:** Yingying Yang.

**Formal analysis:** Yingying Yang.

**Funding acquisition:** Volker Knoop.

**Investigation:** Yingying Yang, Kira Ritzenhofen, Jessica Otrzonsek, Jingchan Xie.

**Methodology:** Yingying Yang, Kira Ritzenhofen, Jessica Otrzonsek, Jingchan Xie, Mareike Schallenberg-Rüdinger, Volker Knoop.

**Project administration:** Mareike Schallenberg-Rüdinger, Volker Knoop.

**Resources:** Volker Knoop.

**Software:** Yingying Yang.

**Supervision:** Mareike Schallenberg-Rüdinger, Volker Knoop.

**Validation:** Yingying Yang, Kira Ritzenhofen, Jessica Otrzonsek, Jingchan Xie, Mareike Schallenberg-Rüdinger.

**Visualization:** Yingying Yang, Kira Ritzenhofen, Volker Knoop.

**Writing – original draft:** Volker Knoop.

**Writing – review & editing:** Yingying Yang, Kira Ritzenhofen, Jingchan Xie, Mareike Schallenberg-Rüdinger, Volker Knoop.

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
