## [Decision Letter · Decision Letter 0]

24 May 2023

Dear Drs. Knoop and Schallenberg-Rüdinger,

Thank you very much for submitting your Research Article entitled 'Beyond a PPR-RNA recognition code: Many aspects matter for the multi-targeting properties of RNA editing factor PPR56' to PLOS Genetics.

The manuscript was fully evaluated at the editorial level and by three independent peer reviewers. The reviewers appreciated the attention to an important topic but identified some concerns that we ask you address in a revised manuscript.

We therefore ask you to modify the manuscript according to the review recommendations. Your revisions should address the specific points made by each reviewer.

Yours sincerely,

Bao-Cai Tan

Guest Editor

PLOS Genetics

Li-Jia Qu

Section Editor

PLOS Genetics

Dear Drs. Knoop and Schallenberg-Rüdinger,

Thanks for submitting your manuscript to PLoS Genetics.

This manuscript by Yang et al. used a heterologous system in E. coli to dissect the determinants in the P. patens PPR56 recognition of its targets by introducing various mutations to both the RNA and the PPR protein. The findings, as pointed by the three reviewers in this field, have a significant contribution to the understanding of the molecular basis of RNA metabolism / RNA editing. Although all the reviewers recognize the high quality and value of this work, they also made some comments and suggestions, which I think could improve this work. In particular,

1) if the authors could analyze function of the E1 and E2 motif in target site recognition, it will provide a complete story.

2) Also, it would strengthen the conclusion if the authors could test the mutation u-16c that improves the conceptual fit to PPR S-13NS.

3) Reviewers also suggested a revision of the text, to make it more clear, logical, and easier to the reader.

4) The parts focusing on “Exploring novel candidate targets” in the Results and Discussion parts are not entirely clear. This should be explained in detail.

Considering these comments and suggestions, a minor revision of this manuscript is required before it can be accepted for publication in PLoS Genetics.

Reviewer's Responses to Questions

**Comments to the Authors:**

Reviewer #1: In this report, the authors tested the RNA editing efficiency of the mutations of moss DYW-type RNA editing factor PPR56 on its target variants in the heterologous bacterial system. They found that (1) the mutations in DYW domain of PPR56 show different effects on the editing of two native targets; (2) the two native targets of PPR56 display different nucleotides opposite of its three central L-type PPR motifs; (3) several transcript features beyond the region ultimately targeted by the PLS-type PPR array likely contribute to enhance the editing activity. RNA secondary structures inhibit RNA editing, whereas native sequences further upstream enhance RNA editing. This work expands our understanding of how PPR proteins recognize and edit substrate RNA targets. The paper is well written, easy to read with experiments supporting most of the conclusions. I have only minor comments.

1. The N-terminal PPR motifs of PPR65 appear to have a significant impact on the editing of nad3eU230SL target, as the mutation u-15c abolishes editing completely. It would be nice if the authors test the mutation u-16c that improves the conceptual fit to PPR S-13NS. Increased editing efficiency is a stronger support for the conclusion.

2. It is not clear from the Materials and methods whether the transit peptide was removed from the proteins assayed in bacterial experiments.

3. The authors should explain why the corresponding triple-mutations converting positions -14, -11 and -8 to the identities in the respective other target decrease editing at the nad4 target to 26% but improve editing at the nad3 target to 76%.

4. In Fig.8, a negative control should be included to avoid that the reason for the increase in editing efficiency at nad3eU230SL site is due to the addition of a nucleic acid sequence before the target.

5. P492. “of” should be “or”.

Reviewer #2: see uploaded review

Reviewer #3: The expression of genes in plant organelles requires coordination of multiple RNA processing steps at the post-transcriptional level. This is run by a multitude of nucleus-encoded RNA-binding proteins (RBPs) that control RNA stability, processing, and degradation. PPR proteins play a pivotal role in these critical processing events in land plant mitochondria. In their MS, Yingying Yang et al analyzed the PPR56 factor from Physcomitrium patens in RNA editing. The authors provide with further details, following their progress made by using a heterologous system for RNA editing in the bacterium E. coli system, regarding the RNA editing properties of PPR56 to its two native transcripts in P. patens, nad3eU230SL and nad4eU272SL. Taking the advantage of this powerful system, the authors introduces various mutations to both the RNA binding PPR module, as well as to the deaminase (DYW) domain, to study the basis of RNA recognition and C-to-U substitutions mediated by the PPR56 cofactor. Undoubtedly, the MS and data therein are of high quality and provides with an excellent contribution to our understanding of these important cellular processes. Noteworthy, the authors are also leading experts in these field, where they have a significant contribution to the understanding of the molecular basis of RNA metabolism / RNA editing and the roles played by nuclear-encoded organellar-localized factors in mitochondrial (or plastidial) RNA maturation. I mostly have minor comments to the authors.

Specific comments:

1. The text may benefit from some revisions to make the MS, data and points raised therein easier to understand and read.

2. Introduction part: Although referred later in this part, it would be easier for the reader if the authors provide with some details about PPR proteins as modular RBPs (often site-specific) as well to their "extra" related domains (e.g., E, E+, DYW) that are associated with substrate RNA "editing to the beginning of the Intro.

3. Mutating the PPR and DYW domains: One way to identify and characterize a domain is to find the part of a target protein that has sequence or structural similarities with a template through homology alignment. This is nicely demonstrated in the seq alignment of the nine DYW domains of the P. patens (Fig. S1) but should be also presented by a phylogenetic illustration as well (i.e., how close are these to one another? And what variation exist in other DYW domains in mosses as well as other species). An interesting question is whether swapping different DYW domains (e.g. from PPR65 into PPR56) may influences the editing, assuming that the sequence specificity resides mainly within the PPR repeats…

4. Regarding the effects of aa's substitutions on editing activity: This is done in a heterologous system and thus the effects should be taken with some cautious. Also, the fact that an amino acid was found to be conserved at the nine DYW domains in P. patens doesn't necessarily indicate that it has any importance for the activity or folding of the protein/domain (the phylogenetic analysis may make this more prevalent). The authors may want to comment on possible effects (predictions) of these substitutions on the folding of the DYW domain or the PPR arrays.

5. The parts focusing on “Exploring novel candidate targets” in the Results and Discussion parts are not entirely clear. The expression, folding and activity of the recombinant P.s.PPR56 in E. coli can affect its specificity and activity resulting with various off-targets in E. coli that may not occur under the native condition in the plant mitochondria.

6. Intro line 47: To a large part, the complex editosomes of angiosperms. Please elaborate on (plant mito) editosome(s).

7. Intro lines 56-57: “Physcomitrium patens has a prominent role with its only 13 C-to-U RNA editing sites assigned to nine site-specific RNA editing factors. Please provide with REFs.

8. Line 63: To readers which are not familiar with PPRs, please explain what are PLS-PPRs.

9. Lines 111-115: This part is unclear and thus could be simplified.

10. Results line 118: “ a typical “complete”, and likely evolutionarily ancestral,” looks to me as an unnecessary comment.

11. Line 137: The coding sequence of PPR56 is cloned in fusion with…, consider in-frame to

12. Line 406: Exploring novel candidate targets – “It is important to keep in mind that orthologues of a functionally characterized”. That's probably depends on the homology, and these might rather represent paralogs in such cases?!

13. The discussion part can be shortened and more focused on the data (see comments 3 to 5).

14. Also, and wityh regard to lines 537-541 is PPR56 is expected to be part of an editosome in P. patens, and whether this factor could be associated with additional RBPs in E. coli?! How these situations influence the activities seen in vivo, in vitro or in the heterologous E.coli systems?

**Have all data underlying the figures and results presented in the manuscript been provided?**

Reviewer #1: Yes

Reviewer #2: Yes

Reviewer #3: **No: **The authors indicate that RNAseq data have been deposited in the SRA archive but don't provide with BioProject REF data.

PLOS authors have the option to publish the peer review history of their article (what does this mean?). If published, this will include your full peer review and any attached files.

Reviewer #1: No

Reviewer #2: No

Reviewer #3: No

---

## [Editor Report · Decision Letter 1]

30 Jul 2023

Dear Dr. Knoop and Schallenberg-Rüdinger,

We are pleased to inform you that your manuscript entitled "Beyond a PPR-RNA recognition code: Many aspects matter for the multi-targeting properties of RNA editing factor PPR56" has been editorially accepted for publication in PLOS Genetics. Congratulations!

Yours sincerely,

Bao-Cai Tan

Guest Editor

PLOS Genetics

Li-Jia Qu

Section Editor

PLOS Genetics

Comments from the reviewers (if applicable):

**Data Deposition**

http://datadryad.org/submit?journalID=pgenetics&manu=PGENETICS-D-23-00376R1

**Press Queries**

---

## [Editor Report · Acceptance letter]

16 Aug 2023

PGENETICS-D-23-00376R1 

Beyond a PPR-RNA recognition code: Many aspects matter for the multi-targeting properties of RNA editing factor PPR56 

Dear Dr Schallenberg-Rüdinger, 

We are pleased to inform you that your manuscript entitled "Beyond a PPR-RNA recognition code: Many aspects matter for the multi-targeting properties of RNA editing factor PPR56" has been formally accepted for publication in PLOS Genetics! Your manuscript is now with our production department and you will be notified of the publication date in due course.

With kind regards,

Zsofi Zombor

PLOS Genetics

On behalf of:
